# CodeT5Mix: A Pretrained Mixture of Encoder-decoder Transformers for Code Understanding and Generation

## Abstract

Pretrained language models (LMs) trained on vast source code have achieved prominent progress in a wide range of code intelligence tasks. Despite their success, they either adopt specific types of network architectures (encoder-only or decoder-only) for different downstream tasks or rely on a single architecture (encoder-decoder or UniLM-style encoder) for all tasks. The latter approach usually results in a sub-optimal performance on a subset of tasks. To address these limitations, we propose "CodeT5Mix", a mixture of encoder-decoder Transformers for code where its components can be flexibly combined based on the target tasks during finetuning, while still enjoying the mutual benefits from the joint pretraining. To endow the model with both code understanding and generation capabilities, we pretrain CodeT5Mix using a mixture of denoising, contrastive learning, matching, and Causal Language Modeling (CLM) tasks on large-scale multilingual code corpora in a stage-wise manner. Additionally, we design a weight sharing strategy in decoders except the feedforward layers, which act as task-specific experts to reduce the interference across tasks of various types. We extensively evaluate CodeT5Mix on seven code-related tasks over twenty datasets and show it achieves state-of-the-art (SoTA) performance on most tasks such as text-to-code retrieval, code completion and generation, and math programming. Particularly, we demonstrate that CodeT5Mix can be used as a unified semi-parametric retrieval-augmented generator with SoTA code generation performance.

## 1 Introduction

Language model pretraining (Chen et al., 2021; Wang et al., 2021c; Feng et al., 2020) has recently demonstrated remarkable success in various downstream tasks in the code domain (Husain et al., 2019; Lu et al., 2021; Hendrycks et al., 2021). By pretraining large-scale language models on massive code-based data (e.g. GitHub public data), these models can learn rich contextual representations which can be transferred to related downstream tasks. However, we found that many of the existing models are specifically designed to perform well only in a subset of tasks (e.g. generative-only tasks or retrieval-only tasks). On other tasks, their performance is suboptimal and the models often require substantial modifications to the architectural features or learning objectives.

Existing models have two main limitations. First, current models follow either encoder-only (Feng et al., 2020; Guo et al., 2021) or decoder-only (Chen et al., 2021; Nijkamp et al., 2022) architectures which are suitable for only a subset of tasks. Specifically, encoder-only models are often used to facilitate retrieval-based tasks such as text-to-code retrieval (Lu et al., 2021). For generative tasks such as code generation (Chen et al., 2021; Hendrycks et al., 2021), decoder-only models are more appropriate. Several approaches have adopted encoder-decoder architectures to adapt to multiple types of tasks (Wang et al., 2021c; Ahmad et al., 2021). While these models can achieve good performance overall, they still fail to beat state-of-the-art encoder-only or decoder-only baselines in some tasks, e.g., retrieval and code completion tasks respectively (Guo et al., 2022). Moreover, Li et al. (2022b) observes that encoder-decoder models do not perform well with in-context learning compared to GPT-style models like Codex (Chen et al., 2021) in code synthesis tasks.

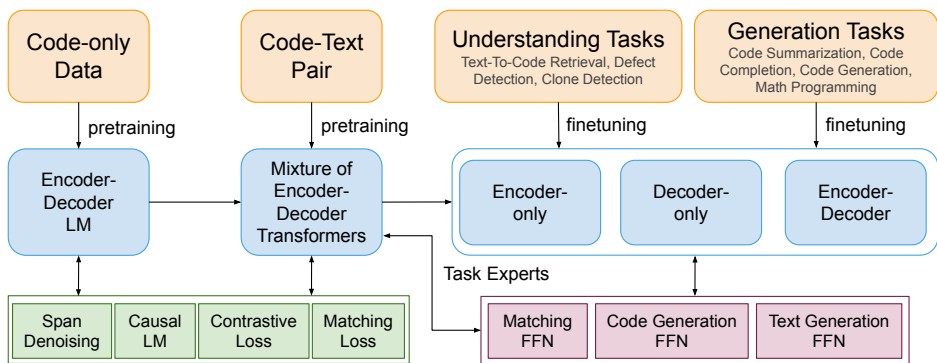

Figure 1: An overview of our CodeT5Mix approach

Secondly, current models are trained in self-supervised learning objectives that might not be appropriate to transfer the models to some downstream tasks. For instance, T5-based models such as (Wang et al., 2021c) are often trained with a span denoising objective. However, in downstream tasks such as code generation (Chen et al., 2021; Hendrycks et al., 2021), most state-of-the-art models are pretrained with a next-token prediction objective which auto-regressively predicts a program token by token. Furthermore, most models do no have specific pretraining tasks to ensure the sharp text/code representation learning which is vital for understanding tasks like text-to-code retrieval. Although recent attempts (Guo et al., 2022) introduce contrastive learning pretraining tasks to cope with this, the performance is still limited by neglecting the fine-grained cross-modal alignments.

To address the above issues, we introduce "CodeT5Mix", a new pretrained language framework for both code understanding and generation (See Fig. 1 for an overview). Specifically, CodeT5Mix includes the following contributions:

- *A mixture of encoder-decoder Transformers*: we introduce a new architectural design for multi-task pretraining and flexible finetuning for both code understanding and generation tasks. CodeT5Mix consists of multimodal encoder and decoder modules, which, in downstream tasks, can be directly repurposed and combined to suit different functionalities.

- *A mixture of self-supervised pretraining tasks*: we adopt a diverse set of pretraining objectives to learn rich representations from both code and text data. We design a stage-wise pretraining strategy to first train on code-only data with span denoising and causal language modeling (CLM) tasks, and then train on text-code data with cross-modal contrastive learning, matching, and CLM tasks, where the matching task is crucial to capture the fine-grained text-code interactions.

- *A weight sharing strategy through task-specific experts*: to optimize multi-task learning while keeping the model parameters affordable, we propose task-specific experts which are designed for different learning tasks while sharing the same backbone contextual representations.

- *A unified model for semi-parametric retrieval-augmented generation*: as CodeT5Mix is capable of both retrieval and generation tasks, we demonstrated that it can be seamlessly adopted as a semi-parametric retrieval-augmented generator to achieve SoTA code generation performance.

- *Thorough evaluation and SoTA performance*: our extensive evaluations show that CodeT5Mix yields significant performance gains on most downstream tasks compared to their SoTA baselines, e.g., 8 text-to-code retrieval tasks (+3.16 avg. MRR), 2 line-level code completion tasks (+2.56 avg. exact match), 2 retrieval-augmented code generation tasks (+5.78 avg. BLEU-4).

- *Open source*: implementation code, data, and pretrained models will be made publicly available.

## 2 RELATED WORK

Typically, code-based language models (LMs) can be categorized into three architectures: encoder-only models like CodeBERT (Feng et al., 2020), GraphCodeBERT (Guo et al., 2021), and CodeMVP (Wang et al., 2022), decoder-only models like CodeGPT (Lu et al., 2021), Codex (Chen et al., 2021), InCoder (Fried et al., 2022) and CodeGen (Nijkamp et al., 2022), and encoder-decoder models like

PLBART (Ahmad et al., 2021), CodeT5 (Wang et al., 2021c), SPT-Code (Niu et al., 2022), and NatGen Chakraborty et al. (2022). For encoder-only and decoder-only models, they are often ideal for either understanding tasks such as code retrieval (Husain et al., 2019) or generation tasks such as code synthesis (Chen et al., 2021; Hendrycks et al., 2021) respectively. For encoder-decoder models, they can be easily adapted for both code understanding and generation but do not always achieve better performance (Wang et al., 2021c; Ahmad et al., 2021) than decoder-only or encoder-only models. In this work, we propose a flexible mixture architecture of Transformers that can operate in various modes: encoder-only, decoder-only, and encoder-decoder.

Prior models employ a limited set of pretraining tasks, which might not be appropriate to transfer the models to some downstream tasks. For instance, there are no specific pretraining tasks in CodeT5 (Wang et al., 2021c) to ensure the learning of a sharp code representation that can distinguish code samples of different semantics, leading to a sub-optimal performance on code understanding tasks like code retrieval (Husain et al., 2019). In light of this, we introduce a contrastive learning task to learn better unimodal representation and a matching task to learn richer bimodal representation, which has been shown to be very effective in text-image retrieval tasks (Li et al., 2021). Additionally, encoder-decoder models (Wang et al., 2021c; Ahmad et al., 2021; Niu et al., 2022; Chakraborty et al., 2022) are not ideal for auto-regressive generation tasks like next-line code completion (Lu et al., 2021; Svyatkovskiy et al., 2020b) [1] and program synthesis (Chen et al., 2021), as these models are trained to recover short spans of limited lengths rather than a whole program sequences. Inspired by recent advances in related NLP domains Tay et al. (2022); Soltan et al. (2022), we propose a mixture of span denoising and CLM tasks to improve the model with better causal generation capability.

More related to our work is UniXcoder (Guo et al., 2022) that adopts a UniLM-style (Dong et al., 2019) model to support various tasks. However, UniXcoder attempts to employ a fixed model with fully shared parameters (but different attention masks) to support many different tasks. The model might suffer from inter-task interference, resulting in unstable model optimization and sub-optimal performance. In CodeT5Mix, we employ a partial weight-sharing approach with a mixture of encoder and decoder Transformers, pretrained with a diverse set of learning objectives. Compared to prior work (Wang et al., 2021c; Guo et al., 2022; Wang et al., 2022), CodeT5Mix also does not rely on any engineering features such as abstract syntax tree or identifier information.

## 3 CodeT5Mix: A Mixture of Encoder-decoder Transformers

To develop CodeT5Mix, we extend CodeT5 (Wang et al., 2021c), a code-aware encoder-decoder language model, with a flexible mixture architecture of encoder-decoder Transformers (Sec. 3.1). In this mixture architecture, we pretrain the models with enhanced learning objectives, with a diverse set of self-supervised tasks over two major stages of pretraining with unimodal and multimodal data (Sec. 3.2). Finally, using the pretrained models, different components can be transferred and activated to serve different functionalities for different downstream tasks (Sec. 3.3). An overview of our pretraining and finetuning process can be seen in Fig. 1, and more details in Fig. 2 and Fig. 3.

### 3.1 Model Architecture

CodeT5Mix consists of multiple Transformers-based encoders and decoders. Each of them can operate a specific functionality and in combination, they serve as a unified multi-task model:

- **Bimodal encoder**, which encodes a text or code snippet into a continuous representation through bidirectional self-attention (Vaswani et al., 2017). Similar to BERT (Devlin et al., 2019), we prepend a special token [CLS] to the input and regard its output embedding at the final Transformer layer as the representations of the corresponding input text or code. We further add a linear layer to map the output to 256-dimensional vectors together with the L2 normalization.

- **Bimodal matching decoder**, which aims to predict whether a text and code snippet share the same semantics. Given a code sample, this decoder first passes it to an embedding layer and a causal self-attention layer. The self-attention representations are then passed to a cross-attention

---

[1]Recent work such as (Tabachnyk & Nikolov, 2022; Fried et al., 2022) demonstrated success using encoder-decoder models for infilling-based code completion, in which the context after the cursor (or after the completed code) is employed. Such code completion setting is not the focus of our paper.

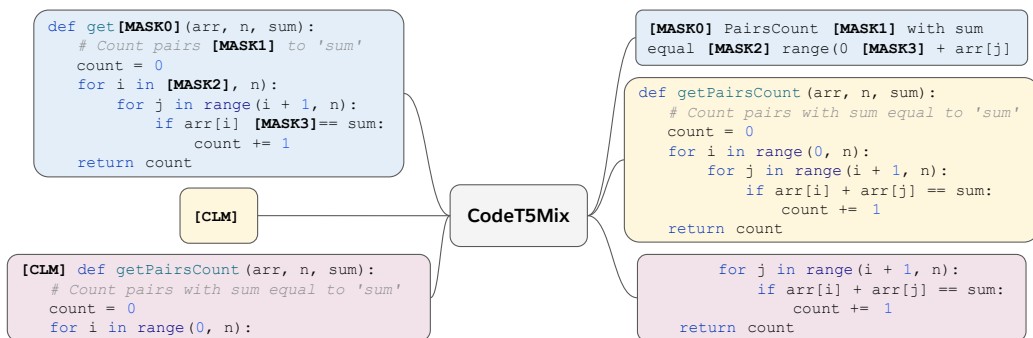

Figure 2: First-stage pretraining objectives of CodeT5Mix on code-only unimodal data

layer which queries relevant signals from the text representations (received from the bimodal encoder). A task-specific [Match] token is prepended to the code input sequence to inform the decoder of the text-code matching functionality, and a [EOS] token is appended to the end of the code input. The output embedding of [EOS] at the last decoder layer is used as the text-code cross-modal alignment representation [2].

- **Unimodal generation decoders**, which can generate an output sequence in programming language/natural language. These decoders follow the same design as the bimodal matching decoder with causal attention and cross-attention layers. When the input is a text sample, we use a code generation decoder and prepend a [CDec] token as the first token in the input sequence. This decoder operates in code generation functionality. When the input is a code sample, we use a text generation decoder and prepend a [TDec] token to the input sequence. This decoder operates in text generation (i.e. code summarization) functionality.

## 3.2 PRETRAINING OBJECTIVES

Given the above model architecture, we develop both understanding and generation-based self-supervised tasks over two major pretraining stages. In the first stage, we pretrain a vanilla encoder-decoder Transformer with massive code-only data using computationally efficient objectives (see Fig. 2). In the second stage, we use the prior model as a backbone to initialize our mixture encoder-decoder Transformers (as described in Sec. 3.1) and train it with a smaller set of code-text data with cross-modal learning objectives (see Fig. 3). We found that this stage-wise training approach can efficiently expose our models to more diverse data to learn rich contextual representations. Note that for each stage, we jointly optimize the model with its pretraining objectives with equal weights.

### 3.2.1 PRETRAINING ON CODE-ONLY UNIMODAL DATA

In the first stage, we pretrain CodeT5Mix on large-scale code-only unimodal data, which can be easily obtained from open-source platforms like GitHub. Note that in this stage, we pretrain the model from scratch using a mixture of span denoising and CLM tasks:

**Span Denoising.** Similar to T5 (Raffel et al., 2020), we randomly replace 15% of the tokens into indexed sentinel tokens (like [MASK0]) in the encoder input, and require the decoder to recover them via generating a combination of these spans. We follow CodeT5 to employ the whole-word masking by sampling spans before subword tokenization to avoid masking partial subtokens.

**Causal Language Modeling.** To optimize our model for auto-regressive generation, we introduce two variants of CLM. In the first variant, we randomly select a pivot location and regard the context before it as the source sequence and the sequence after it as the target output. We restrict the pivot location to be uniformly sampled between 10% and 90% of the whole sequence. We prepend a special token [CLM] to the input sequence. The second CLM variant is a decoder-only generation task and can be viewed as an extreme case of the first variant. In this task, we always pass a single [CLM] token to the encoder input and require the decoder to generate the full code sequence.

---

[2]Note that we choose the embedding of [EOS] instead of [Match] as the decoder employs causal self-attention masks and only the last decoder token can attend to all the contexts.

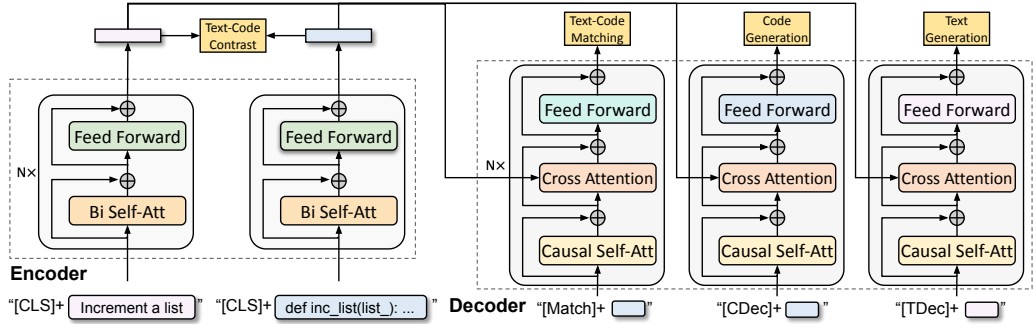

Figure 3: Second-stage pretraining objectives of CodeT5Mix on text-code bimodal data. Model components with the same color have tied weights: 2 encoders are fully shared and 3 decoders except the Feed Forward layers (FNN) are shared. Although this mixture architecture incurs extra parameter cost of two FNN experts compared to a standard encoder-decoder, such cost can be waived in finetuning as only a FFN expert is activated for a task and incurs no extra computational cost.

Compared to the first variant, this task aims to provide more dense supervision signals to train the decoder as an independent full-fledged code generation module.

### 3.2.2 PRETRAINING ON TEXT-CODE BIMODAL DATA

In the second stage, we use text-code bimodal data at the function level curated by Husain et al. (2019). In this setting, each text-code pair is a code function and its corresponding docstring describing its semantics. Such a bimodal data format facilitates the exploration of cross-modal retrieval and generation tasks. Note that we adopt the CodeT5Mix architecture (see Fig. 3) in which the encoder and decoder components are initialized from the checkpoints after the first pretraining stage.

**Text-Code Contrastive Learning.** This task aims to align the feature space of text and code encoder by pulling together the representations of positive text-code pairs and pulling apart the negative pairs. This task has been shown to be effective for code understanding (Guo et al., 2022). This task only activates the bimodal encoder to produce a text/code embedding.

To enrich the negative samples, we use a momentum encoder to store embeddings of samples from previous mini-batches, as similarly adopted by He et al. (2020); Li et al. (2022a). Specifically, the momentum encoder maintains a queuing system that enqueues the samples in the current mini-batch and dequeues the samples in the oldest mini-batch. To ensure the consistency of representations across training steps, we update the momentum encoder by linear interpolation of the original encoder and the momentum encoder. Besides, since text and code samples might be loosely paired and each text/code sample can have multiple positive pairs, we also use the momentum encoder to create soft labels and consider the potential positives in the negative pairs.

**Text-Code Matching.** This task activates the bimodal matching decoder, which aims to learn better bimodal representations that capture the fine-grained alignment between text and code modalities. We build a linear layer on top of the output embedding of the decoder for a binary classification task, which predicts whether a text-code pair is positive (matched) or negative (unmatched).

In order to find more informative negatives, we employ a hard negative mining strategy (Li et al., 2021). Specifically, we sample hard negatives based on the contrastive-based similarity scores between the current sample and previous samples in the queue maintained by the momentum encoder. As such, harder negatives are more likely to be selected. For a batch of positive pairs, we construct two batches of negative pairs by mining negatives from the text/code queue with a code/text query.

**Text-Code Dual Generation.** This task focuses on a cross-modal generative objective between text and code through a dual multimodal conversion: text-to-code generation and code-to-text generation (i.e. code summarization). Each conversion separately activates the corresponding (unimodal) code/text generation decoder. This task is an effective learning objective to close the gap between the pretraining and finetuning stage on generation-based downstream tasks (Wang et al., 2021c).

### 3.2.3 Efficient Weight-sharing Strategy with Task-specific Experts

While we only use a single encoder to process bimodal data, we adopt a partially sharing scheme among the three decoders. Specifically, among the decoders, we share the parameters of the self-attention and cross-attention layers. We found that the contextual representations learned from these layers should be shared across text-code matching and text-code dual generation tasks, enabling cross-task generalization at the context level. Intuitively, these multimodal tasks benefit from sharing contextual representations, e.g., a text-to-code generation decoder could benefit from semantic-aware code representations well learned from a text-code matching task.

While the contextual representations can be learned across tasks, we deploy different feed-forward layers with separate weights, similar to the modality-specific experts in vision-language encoders by Wang et al. (2021a). These layers acts as task-specific experts in order to reduce the task interference during pretraining. Another benefit of this approach is keeping the model parameters affordable, even with a mixture architecture like CodeT5Mix. Note that a weight-sharing scheme that fully shares all parameters among the decoders can save more parameter costs, but will result in serious interference and performance drops in downstream tasks (see Sec. 4.4). Moreover, the extra parameter cost of the introduced task experts can be waived during finetuning, as only a single task expert will be activated for one downstream task, thereby incurring no extra computational cost.

### 3.3 Finetuning on Downstream Tasks

**Understanding Tasks.** We employ three tasks to evaluate the code understanding capability of our models: text-to-code retrieval, defect detection, and clone detection. CodeT5Mix can support these understanding tasks in two ways: first, it employs the bimodal encoder to obtain text/code embeddings, which can be either passed to a binary classifier for defect detection, or used to compute a similarity score for clone detection and code retrieval tasks; secondly, CodeT5Mix can additionally use the text-code matching decoder to predict the matching probabilities.

**Decoder-only Tasks.** We use next-line code completion tasks to evaluate the decoder-only generation capability of CodeT5Mix. In this setting, we always feed a `[CLM]` token to the encoder input and pass the source sequence to the decoder. The decoder is required to understand the given context and complete the next line of code. We freeze the weights of the encoder and the cross-attention layers at the decoder to reduce the number of trainable parameters for efficient training.

**Seq2Seq Generation Tasks.** CodeT5Mix can be transferred to various sequence-to-sequence (Seq2Seq) generation tasks such as code summarization, code generation, and math programming. For math programming tasks, as their domains are much different from our text-code pretraining data, we employ models pretrained from the first pretraining stage for evaluation.

## 4 Experiments

We conduct comprehensive experiments on seven code understanding and generation tasks across nine programming languages (PLs). Detailed experimental setups can be found in the Appendix.

**Baselines.** We develop two variants of CodeT5Mix: base (220M) and large (770M) models. We compare CodeT5Mix with state-of-the-art code-based pretrained LMs. For **encoder-only** models, we consider RoBERTa (Liu et al., 2019), CodeBERT (Feng et al., 2020) trained with masked language modeling, GraphCodeBERT (Guo et al., 2021) using data flow extracted from abstract syntax tree (AST) of code, and SYNCOBERT (Wang et al., 2021b) that incorporates AST and contrastive learning. For **decoder-only** models, we employ GPT-2 (Radford et al., 2019) and CodeGPT (Lu et al., 2021), where both are pretrained using a CLM objective. For **encoder-decoder** models, we consider PLBART (Ahmad et al., 2021) and CodeT5 (Wang et al., 2021c) that employ a unified framework to support both understanding and generation tasks. We further include CodeT5-large results from Le et al. (2022). We also compare another unified model UniXcoder (Guo et al., 2022) that employs UniLM-style masking (Dong et al., 2019). For model sizes, CodeBERT, GraphCodeBERT, SYNCOBERT, UniXcoder are based on RoBERTa-base and have 125M parameters, while GPT-2 and CodeGPT has 124M and PLBART has 140M. Notably, CodeT5Mix only employs roughly half of its parameters when acted in encoder-only and decoder-only modes.

## 4.1 EVALUATION ON UNDERSTANDING TASKS

**Text-to-Code Retrieval.** This task aims to find the most semantically related code snippet at the function level from a collection of candidate codes based on a natural language query. We consider three datasets for evaluation: CodeSearch-Net (Husain et al., 2019),

Table 1: Text-to-Code Retrieval results (MRR) on CodeXGLUE

| Model | CodeSearchNet | | | | | | | CosQA | AdvTest |
| | Ruby | JS | Go | Python | Java | PHP | Overall | | |
|---|---|---|---|---|---|---|---|---|---|
| RoBERTa | 58.7 | 51.7 | 85.0 | 58.7 | 59.9 | 56.0 | 61.7 | 60.3 | 18.3 |
| CodeBERT | 67.9 | 62.0 | 88.2 | 67.2 | 67.6 | 62.8 | 69.3 | 65.7 | 27.2 |
| GraphCodeBERT | 70.3 | 64.4 | 89.7 | 69.2 | 69.1 | 64.9 | 71.3 | 68.4 | 35.2 |
| SYNCOBERT | 72.2 | 67.7 | 91.3 | 72.4 | 72.3 | 67.8 | 74.0 | - | 38.3 |
| PLBART | 67.5 | 61.6 | 88.7 | 66.3 | 66.3 | 61.1 | 68.6 | 65.0 | 34.7 |
| CodeT5-base | 71.9 | 65.5 | 88.8 | 69.8 | 68.6 | 64.5 | 71.5 | 67.8 | 39.3 |
| UniXcoder | 74.0 | 68.4 | 91.5 | 72.0 | 72.6 | 67.6 | 74.4 | 70.1 | 41.3 |
| CodeT5Mix-base | 77.7 | 70.8 | 92.4 | 75.6 | 76.1 | 69.8 | 77.1 | 72.7 | 43.3 |
| CodeT5Mix-large | **78.0** | **71.3** | **92.7** | **75.8** | **76.2** | **70.1** | **77.4** | **74.0** | **44.7** |

CosQA (Huang et al., 2021), and AdvTest (Lu et al., 2021), which are curated from the original CodeSearchNet by filtering data with low-quality queries, adopting real-world queries from a modern search engine, and obfuscating identifiers to normalize the code. In this task, we activate the encoder and matching decoder of CodeT5Mix and use Mean Reciprocal Rank (MRR) as the metric.

From Table 1, our CodeT5Mix-base significantly outperforms all existing encoder-only and encoder-decoder models and our large variant further sets new SoTA results, surpassing the previous SoTA UniXcoder by more than 3 absolute MRR points on all 3 tasks across 8 datasets. This implies CodeT5Mix is a robust code retriever model to handle queries with diverse formats and PLs. Besides, CodeT5Mix-base yields substantial performance gains over CodeT5-base, which can be attributed to the text-code contrastive learning and matching objectives that facilitate better unimodal and bimodal representation learning. Particularly, compared to prior contrastive-learning approaches like SYNCOBERT and UniXcoder, our models achieve much better results, which can be attributed to our bimodal matching decoder that allows for more fine-grained text-code alignments.

**Defect Detection and Clone Detection.** Defect detection is to predict whether a code is vulnerable to software systems or not, while clone detection aims to measure the similarity between two code snippets and predict whether they have a common functionality. We use benchmarks from CodeXGLUE (Lu et al., 2021) and use accuracy and F1 score as the metrics. In Table 2, we can see CodeT5Mix models achieve new SoTA accuracy on defect detection. On clone detection, our model achieves comparable results to SoTA models, where the performance increase tend to be saturated.

Table 2: Results on understanding tasks: code defect detection and clone detection

| Model | Defect | Clone Detection | | |
| | Acc | Rec | Prec | F1 |
|---|---|---|---|---|
| RoBERTa | 61.1 | 95.1 | 87.8 | 91.3 |
| CodeBERT | 62.1 | 94.7 | 93.4 | 94.1 |
| GraphCodeBERT | - | 94.8 | 95.2 | 95.0 |
| PLBART | 63.2 | 94.8 | 92.5 | 93.6 |
| CodeT5-base | 65.8 | 95.1 | 94.9 | 95.0 |
| UniXcoder | - | 92.9 | **97.6** | **95.2** |
| CodeT5Mix-base | 66.1 | 96.4 | 94.1 | **95.2** |
| CodeT5Mix-large | **66.7** | **96.7** | 93.5 | 95.1 |

## 4.2 EVALUATION ON CODE COMPLETION TASKS

We evaluate the decoder-only generation capability of CodeT5Mix through a line-level code completion task, which aims to complete the next whole line code based on the previous code contexts. We employ PY150 (Raychev et al., 2016) and GitHub JavaCorpus (Allamanis & Sutton, 2013) from CodeXGLUE, and use exact match (EM) accuracy and Levenshtein edit similarity (Svyatkovskiy et al., 2020a) as evaluation metrics. Typically, this task requires a decoder-only model for efficient training.

Table 3: Results on line-level code completion

| Model | PY150 | | JavaCorpus | |
| | EM | Edit Sim | EM | Edit Sim |
|---|---|---|---|---|
| Transformer | 38.51 | 69.01 | 17.00 | 50.23 |
| GPT-2 | 41.73 | 70.60 | 27.50 | 60.36 |
| CodeGPT | 42.37 | 71.59 | 30.60 | 63.45 |
| PLBART | 38.01 | 68.46 | 26.97 | 61.59 |
| CodeT5-base | 36.97 | 67.12 | 24.80 | 58.31 |
| UniXcoder | 43.12 | 72.00 | 32.90 | 65.78 |
| CodeT5Mix-base | 43.42 | 73.69 | 34.23 | 68.75 |
| CodeT5Mix-large | **44.86** | **74.22** | **36.27** | **70.33** |

As shown in Table 3, our CodeT5Mix achieves new SoTA results compared to both decoder-only and encoder-decoder models in both metrics. In particular, CodeT5Mix-base yields substantial improvements over CodeT5-base by 6.45 and 9.43 EM scores on PY150 and JavaCorpus respectively. This is mainly due to the causal language modeling task introduced in our first-stage pretraining, which allows the decoder to see longer sequences instead of a combination of discrete spans in CodeT5, leading to a better causal sequence generation capability.

Table 5: Results on math programming tasks. [†] denotes few-shot learning.

(a) Results on MathQA-Python

| Model | Pass@80 |
|---|---|
| Codex Davinci[†] | 42 |
| LaMDA 8B | 74.7 |
| LaMDA 64B | 79.5 |
| LaMDA 137B | 81.2 |
| GPT-Neo+sampling 125M | 84.7 |
| CodeT5-base 220M | 71.5 |
| CodeT5-large 770M | 72.3 |
| CodeT5Mix-base 220M | 85.6 |
| CodeT5Mix-large 770M | **87.4** |

(b) Results on GSM8K-Python

| Model | Pass@1 | Pass@100 |
|---|---|---|
| OpenAI 6B | 21.8 | 70.9 |
| Codex Davinci[†] | 17 | 71 |
| LaMDA 137B[†] | 7.6 | - |
| PaLM-Coder 540B[†] | **50.9** | - |
| GPT-Neo+sampling 2.7B | 19.5 | 41.4 |
| CodeT5-base 220M | 13.5 | 58.4 |
| CodeT5-large 770M | 19.3 | 61.4 |
| CodeT5Mix-base 220M | 22.4 | 70.5 |
| CodeT5Mix-large 770M | 26.2 | **73.8** |

## 4.3 EVALUATION ON SEQ2SEQ GENERATION TASKS

**Code Summarization and Code Generation.** Code summarization aims to summarize a code snippet into docstrings while code generation is to produce a function based on a natural language description. We employ the clean version of CodeSearchNet dataset in six PLs for code summarization and a Java ConCode (Iyer et al., 2018) for code generation. For evaluation metric, we employ BLEU-4 (B4), exact match (EM) accuracy, and CodeBLEU (CB) (Ren et al., 2020) which accounts for syntactic and semantic matches based on the code structure in addition to the n-gram match.

From CodeT5Mix, we activate the encoder with the text generation decoder for code summarization, and with the code generation decoder for code generation. We report the results in Table 4.

Overall, we found that encoder-decoder models (CodeT5 and CodeT5Mix) generally outperform both encoder-only and decoder-only models, as well as the unified UniXcoder with controlled masks on both tasks. This implies that encoder-decoder models can better support Seq2Seq generation tasks. Our CodeT5Mix-large achieves new SoTA results on both tasks across various metrics.

Table 4: Results on code summarization (average BLEU-4 over 6 PLs) and code generation

| Model | Sum. | Generation | | |
|---|---|---|---|---|
| | B4 | EM | B4 | CB |
| CodeBERT | 17.83 | - | - | - |
| CodeGPT | - | 20.10 | 32.79 | 35.98 |
| PLBART | 18.32 | 18.75 | 36.69 | 38.52 |
| CoTexT | 18.54 | 20.10 | 37.40 | 40.14 |
| UniXcoder | 19.30 | 22.60 | 38.23 | - |
| CodeT5-base | 19.55 | 22.30 | 40.73 | 43.20 |
| CodeT5-large | 19.87 | 22.65 | 42.66 | 45.08 |
| CodeT5Mix-base | 19.81 | 22.30 | 41.23 | 44.03 |
| CodeT5Mix-large | **20.15** | **22.75** | **42.73** | **45.56** |

**Math Programming.** To evaluate models for code generation, exact match or BLEU scores might be limited as there can be multiple forms of correct program solutions. Besides, Chen et al. (2021) found that the functional correctness of generated codes correlates poorly with their BLEU scores. Therefore, we further consider two math programming tasks, namely MathQA-Python (Austin et al., 2021) and GSM8K-Python (Cobbe et al., 2021), where code correctness can be measured based on the execution outputs of code programs. The task is to generate Python programs to solve mathematical problems described in natural language descriptions. We employ *pass@k*, which measures the percentage of problems solved using $k$ generated programs.

Apart from CodeT5, we compare with very large-scale decoder-only models including Codex (Chen et al., 2021), LaMDA (Austin et al., 2021), PaLM-Coder (Chowdhery et al., 2022), and GPT-Neo (Black et al.) with self-sampled learning strategy (Ni et al., 2022). As shown in Tables 5a and 5b, we found that CodeT5Mix achieves significant performance gains, outperforming many pretrained models of much larger sizes. Specifically, our CodeT5Mix-large achieves new SoTA results of 87.4 *pass@80* on MathQA-Python and 73.8 *pass@100* on GSM8K-Python. On GSM8K-Python, Our model achieves the second best result of 26.2 *pass@1*, only behind PaLM-Coder which has a much larger size than CodeT5Mix (at 540B) and was exposed to much larger pretraining data.

## 4.4 ABLATION STUDY

We conduct an ablation study to analyze the impacts of different architectural designs (removing either the matching decoder or both code and text generation decoder) and weight-sharing strategies (fully separate or shared decoders compared to shared decoders except FFNs). Table 6 reports the results on three representative tasks, namely text-to-code retrieval (MRR), code summarization (BLEU) and generation (CodeBLEU). Note that for code retrieval and summarization that share the same dataset source of clean CodeSearchNet (Lu et al., 2021), we use a CodeT5Mix-base checkpoint

Table 6: Ablation results of CodeT5Mix with varying model designs and weight sharing strategies.

| Model | Text-to-Code Retrieval | | | | | | | Summarization | Generation |
| | Ruby | JavaScript | Go | Python | Java | PHP | Overall | | |
|---|---|---|---|---|---|---|---|---|---|
| CodeT5Mix-base | 75.85 | 69.91 | 91.33 | 74.45 | 75.28 | 68.22 | 75.84 | 19.53 | 44.03 |
| no match decoder | 74.51 | 69.09 | 90.69 | 71.81 | 71.81 | 67.70 | 74.19 | 19.49 | 43.76 |
| no code/text decoder | 74.39 | 68.68 | 91.36 | 73.46 | 74.60 | 67.71 | 75.03 | - | - |
| separate decoder | 74.72 | 69.46 | 91.45 | 73.85 | 74.55 | 67.60 | 75.27 | 19.38 | 43.63 |
| shared decoder | 75.84 | 69.68 | 91.07 | 73.38 | 74.31 | 67.43 | 75.29 | 19.24 | 43.44 |

that was jointly trained on all six PLs in a multitask setting for evaluation to remove the impact of task-specific finetuning. For the code generation task, we report the task-specific finetuning results.

Overall, in CodeT5Mix, we found that all components of our mixture architecture can complement to each other and lead to the best results on all tasks. This is observed by a performance drop by removing the matching decoder or code/text generation decoder. Particularly, we found that the matching decoder is critical to text-to-code retrieval performance (1.65 average MRR over 6 datasets dropped by removing it), as this decoder can learn a better bimodal representation that captures the fine-grained alignment between text and code, while for code summarization and generation tasks it has only slight benefits. For the weight sharing strategy, we found that our proposed task-specific FFN experts allow the model to balance various tasks with insignificant extra overhead of FFN parameters. This is demonstrated by the consistent performance gains over the model variants (the last two rows in Table 6) with a fully separate decoder or shared decoder design.

## 4.5 RETRIEVAL-AUGMENTED CODE GENERATION

As our model is capable of both code retrieval and generation, it can be naturally exploited as a unified semi-parametric retrieval-augmented generator. To explore such setting, we follow Parvez et al. (2021) to evaluate two code generation tasks by reversing the input and output order of code summarization on Java and Python and using their released deduplicated retrieval codebase. We evaluate our model in three settings: retrieval-based, generative, and retrieval-augmented (RA) generative.

Table 7: Results of retrieval-augmented code generation

| Model | Java | | | Python | | |
| | EM | B4 | CB | EM | B4 | CB |
|---|---|---|---|---|---|---|
| | Retrieval-based | | | | | |
| BM25 | 0.00 | 4.90 | 16.00 | 0.00 | 6.63 | 13.49 |
| SCODE-R | 0.00 | 25.34 | 26.68 | 0.00 | 22.75 | 23.92 |
| CodeT5Mix-base | 0.00 | 28.74 | 31.00 | 0.00 | 27.30 | 26.51 |
| | Generative | | | | | |
| CodeBERT | 0.00 | 8.38 | 14.52 | 0.00 | 4.06 | 10.42 |
| GraphCodeBERT | 0.00 | 7.86 | 14.53 | 0.00 | 3.97 | 10.55 |
| CodeGPT | 0.00 | 7.10 | 14.90 | 0.01 | 3.11 | 11.31 |
| PLBART | 0.00 | 10.10 | 14.96 | 0.00 | 4.89 | 12.01 |
| CodeT5Mix-base | 0.00 | 10.33 | 20.54 | 0.00 | 4.40 | 13.88 |
| | Retrieval Augmented Generative | | | | | |
| REDCODER-EXT | 10.21 | 28.98 | 33.18 | 9.61 | 24.43 | 30.21 |
| CodeT5Mix-base | **11.66** | **33.83** | **40.60** | **11.83** | **31.14** | **36.39** |

For the retrieval-based setting, we activate our encoder to retrieve the top-1 code sample as the prediction given a text query, while for the RA generative setting, we append the combination of top-$k$ retrieved samples ($k$=1 in our work) to the input and activate the code generation decoder.

As shown in Table 7, we found that our CodeT5Mix achieves significantly better results in all categories, especially in the retrieval-based and RA generative setting. Compared to the previous SoTA model of REDCODER-EXT (Parvez et al., 2021) that separately employs GraphCodeBERT as the retriever and PLBART as the generator, our model can seamlessly operate as a better code retriever and generator, leading to superior retrieval-augmented generative performance.

## 5 CONCLUSION

We present CodeT5Mix, a new code-based pretrained model with a mixture of encoder-decoder Transformer modules that can be flexibly combined for a wide range of code understanding and generation tasks. To train CodeT5Mix, we introduce a diverse set of pretraining tasks including denoising, causal language modeling, contrastive learning and matching to learn rich representations from both code-only data and bimodal code-text data. We design an efficient weight-sharing strategy with task-specific FFN experts to reduce inter-task interference while keeping the model parameters affordable. Extensive experiments on seven downstream tasks verify the superiority of our model. We further showcase that CodeT5Mix can fully support a semi-parametric retrieval-augmented code generation approach as the model can effectively operate in both code retrieval and generation tasks.

## 6 ETHICS STATEMENT

Improving code understanding and generation systems has the potential to create substantial positive impacts on the society, by making programming more accessible and developers more productive via natural language interfaces. However, as discussed in detail by Chen et al. (2021), several ethical considerations have to be made when deploying such systems at scale. One such limitation is the risk of generated code summaries or comments containing toxic and insensitive language. Several studies have discussed the problem of ensuring non-toxic natural language generation with some solutions ranging from the use of RL (Ouyang et al., 2022), weighted decoding (Krause et al., 2021) to safety-specific control tokens (Xu et al., 2020).

Besides safety, substantial consideration is also needed to account for the broader intellectual property implications of code generation and search systems before deployment. For instance, with code search, appropriate attribution should be provided to the source along with the retrieved results. Code generated from deep learning models can also contain security vulnerabilities, requiring expert review before adoption.

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

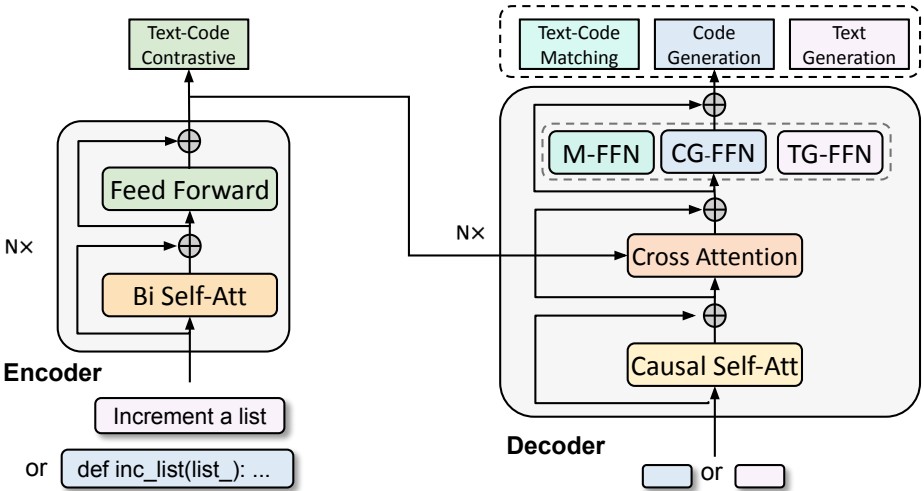

Figure 4: Another view of CodeT5Mix as an encoder-decoder model with task-specific FFN experts: M-FFN for text-code matching loss, TG-FFN for text generation, and CG-FFN for code generation.

## A  FURTHER EXPLANATION OF CODET5MIX

### A.1  ANOTHER VIEW OF CODET5MIX

The mixture architecture of CodeT5Mix can be interpreted from another view as a unified encoder-decoder with task-specific FFN experts in a shared decoder, which is illustrated in Fig. 4. Note that while we introduce extra parameter cost of two FFN layers compared to a standard encoder-decoder model during pretraining, only one FFN layer is activated when finetuning on one specific downstream task, thereby incurring no additional computational cost in fact.

### A.2  TEXT-CODE CONTRASTIVE LEARNING AND MATCHING PRETRAINING TASK

To expose the model on more diverse set of pretraining data, we employ a stage-wise pretraining process to first train CodeT5Mix on large-scale code-only data with span denoising and causal language modeling (CLM) tasks, then train on smaller set of text-code bimodel data using text-code contrastive learning, matching, and CLM tasks. Below, we provide detailed formulas for text-code contrastive learning and matching tasks at the second-stage pretraining on text-code pairs.

**Text-Code Contrastive Learning** activates the bimodal encoder to learn better unimodal (text/code) representations by computing a similarity score such that parallel text-code pairs have higher scores. Given a text T and a code C, we first learn representations $\mathbf{h}^t$ for text $T$ and $\mathbf{h}^c$ for code $C$ by mapping the `[CLS]` embeddings to normalized lower-dimensional (256-d) representations from the bimodal encoder. Given a batch of $N$ text-code pairs, we obtain text vectors $\{\mathbf{h}^t\}_{i=1}^N$ and code vectors $\{\mathbf{h}^c\}_{i=1}^N$ to compute text-to-code and code-to-text and similarities:

$$s_{i,j}^{t2c} = \mathbf{h}_i^{t\top}\mathbf{h}_j^c, s_{i,j}^{c2t} = \mathbf{h}_i^{c\top}\mathbf{h}_j^t \tag{1}$$

$$p_i^{t2c}(T) = \frac{\exp\left(s_{i,i}^{t2c}/\tau\right)}{\sum_{j=1}^N \exp\left(s_{i,j}^{t2c}/\tau\right)}, p_i^{c2t}(C) = \frac{\exp\left(s_{i,i}^{c2t}/\tau\right)}{\sum_{j=1}^N \exp\left(s_{i,j}^{c2t}/\tau\right)} \tag{2}$$

where $s_{i,j}^{t2c}$ represents text-to-code similarity of text of $i$-th pair and code of $j$-th pair, and $s_{i,j}^{c2t}$ is the code-to-text similarity, $\tau$ is learned temperature parameter. $p_i^{t2c}(T)$ and $p_i^{c2t}(C)$ are the softmax-normalized text-to-code and code-to-text similarities for the $i$-th text and code.

Let $\mathbf{y}^{t2c}(T)$ and $\mathbf{y}^{c2t}(C)$ denote the ground-truth one-hot similarity, where negative pairs have a probability of 0 and the positive pair has a probability of 1. The text-code contrastive loss from a

corpus $D$ of text-code pairs is defined as the cross-entropy H between $\mathbf{p}$ and $\mathbf{y}$:

$$\mathcal{L}_{contra} = \frac{1}{2}\mathbb{E}_{(T,C)\sim D}[H(\mathbf{y}^{t2c}(T), \mathbf{p}^{t2c}(T)) + H(\mathbf{y}^{c2t}(C), \mathbf{p}^{c2t}(C))] \tag{3}$$

**Text-Code Matching**  activates the bimodal matching decoder to predict whether a pair of text and code is positive (matched) or negative (unmatched). We employ the output embedding of the [EOS] token as the fused bimodal representation for a text-code pair $(T, C)$. Followed by a linear layer and softmax, we compute a two-class probability $p^{match}(T, C)$ and define the matching loss:

$$\mathcal{L}_{match} = \mathbb{E}_{(T,C)\sim D}[H(\mathbf{y}^{match}(T, C), \mathbf{p}^{match}(T, C))] \tag{4}$$

where $\mathbf{y}^{match}(T, C)$ is a 2-dimensional one-hot vector representing the ground-truth label.

## B  PRETRAINING

### B.1  PRETRAINING DATASET

We enlarge the pretraining dataset of CodeSearchNet (Husain et al., 2019) with the recently released GitHub Code dataset[3]. We select nine PLs (Python, Java, Ruby, JavaScript, Go, PHP, C, C++, C#) and filter the dataset by preserving only permissively licensed code[4] and files with 50 to 2000 tokens. Besides, we filter out the overlapped subset with CodeSearchNet and other downstream tasks by checking their GitHub repositories. Note that although we employ the duplicated data version in which duplicates are filtered out based on the exact match ignoring the whitespaces, there might be some potential remaining duplicates. However, we do not expect any remaining duplication will impact our model performance significantly. We use the CodeT5 tokenizer and the resulting multilingual dataset has 51.5B tokens, more than 50x larger than CodeSearchNet.

Table 8: Data statistics of both unimodal and bimodal pretraining data

| Dataset | Language | # Sample | Total size |
|---|---|---|---|
| Ours | Ruby | 2,119,741 | |
| | JavaScript | 5,856,984 | |
| | Go | 1,501,673 | |
| | Python | 3,418,376 | |
| | Java | 10,851,759 | 37,274,876 files |
| | PHP | 4,386,876 | |
| | C | 4,187,467 | |
| | C++ | 2,951,945 | |
| | C# | 4,119,796 | |
| CodeSearchNet | Ruby | 49,009 | |
| | JavaScript | 125,166 | |
| | Go | 319,132 | 1,929,817 text-code |
| | Python | 453,772 | pairs at function level |
| | Java | 457,381 | |
| | PHP | 525,357 | |

We report the data statistics of both code-only and text-code pretraining datasets in Table 8. From the table, we can see that our curated dataset from GitHub code has a much larger data size at the file level than the CodeSearchNet bimodal data at the function level, allowing our model to learn rich representation at the first stage of pretraining. Different from CodeT5 (Wang et al., 2021c) that employs both unimodal and bimodal data in CodeSearchNet Husain et al. (2019), we only employ its bimodal subset for the second stage pretraining of our CodeT5Mix, which aims to adapt our model to better support text-code related tasks like text-to-code retrieval and generation. For data tokenization, we employ the code-specific subword tokenizer as described by Wang et al. (2021c).

---

[3]https://huggingface.co/datasets/codeparrot/github-code
[4]Permissive licenses: "mit", "apache-2", "bsd-3-clause", "bsd-2-clause", "cc0-1.0", "unlicense", "isc"

Table 9: Data statistics of the CSN dataset (Husain et al., 2019)

| Language | Training | Validation | Test | #Candidates |
|---|---|---|---|---|
| Go | 167,288 | 7,325 | 8,122 | 28,120 |
| Java | 164,923 | 5,183 | 10,955 | 40,347 |
| JavaScript | 58,025 | 3,885 | 3,291 | 13,981 |
| PHP | 241,241 | 12,982 | 14,014 | 52,660 |
| Python | 251,820 | 13,914 | 14,918 | 43,827 |
| Ruby | 24,927 | 1,400 | 1,261 | 4,360 |

## B.2 PRETRAINING DETAILS

We pretrained a CodeT5Mix-base (220M) and CodeT5Mix-large (770M) from scratch following T5's architecture (Raffel et al., 2020) on a cluster with 16 A100-40G GPUs on Google Cloud Platform. Note that we introduce task-specific FFN experts in decoders, where each FFN expert incurs insignificant extra parameter costs of 28M and 101M for the base and large model respectively. These insignificant extra parameter costs can be waived when applied on a specific task as we only activate one task-specific FFN expert. We adopt a stage-wise pretraining process to pretrain CodeT5Mix first on the large-scale unimodal dataset and then on the smaller bimodal dataset.

In the first stage, we warm up the model with the span denoising task for $10k$ training steps, and then joint training with another two CLM tasks with equal probability for $100k$ steps. We employ a linear decay learning rate (LR) scheduler with a peak learning rate of 2e-4 and set the batch size to 2048 for denoising and 512 for CLM. To prepare the input and output data, we set the maximum length to 512 for the denoising task, and set the maximum length to 768 and 600 for source and target sequences for the code completion CLM, 1 and 1024 for the decoder-only generation CLM.

In the second stage, we jointly optimize four losses of contrastive learning, matching, and two CLM losses with equal weights for 10 epochs with a batch size of 256. We employ a peak learning rate of 1e-4 and set the maximum sequence length to 420 and 128 for code and text sequences. For all experiments, we employ an AdamW optimizer (Loshchilov & Hutter, 2019) with a 0.1 weight decay. We also employ the DeepSpeed's ZeRO Stage 2 (Rasley et al., 2020) with mixed precision training of FP16 for training acceleration.

## C FINETUNING TASKS

### C.1 TEXT-TO-CODE RETRIEVAL

Text-to-code retrieval (or code search), is the task of finding the best code sample that is most relevant to a natural language query, from a collection of code candidates. We experiment CodeT5Mix with three major benchmarks: CodeSearchNet (CSN) (Husain et al., 2019), CosQA (Huang et al., 2021), and AdvTest (Lu et al., 2021). CSN consists of six programming languages in total, and the dataset is curated by filtering low-quality queries through handcrafted rules, following (Guo et al., 2021). For instance, an example handcraft rule is to filter examples in which the number of tokens in query is shorter than 3 or more than 256. The resulting dataset statistics can be seen in Table 9.

CosQA and AdvTest are two related benchmarks that are derived from the CSN data. Specifically, instead of natural language queries, CosQA uses logs from Microsoft Bing search engine as queries, each of which is annotated by 3 human annotators (Huang et al., 2021). AdvTest is created from the Python split of the CSN data but the code samples are normalized with obfuscated variable names to better evaluate the understanding abilities of current models.

For training, we set the maximum sequence to 350 and 64 for code and text. We set the learning rate as 2e-5 and finetune the model for 10 epochs. We employ distributed training using Pytorch DistributedDataParallel[5] on 8 A100s and the total batch size is 64. For momentum encoders, we maintain a separate text/code queue with a size of 57600, and allow the matching decoder to retrieve 64 hard negatives from the queues for hard negative mining.

---

[5]https://pytorch.org/

## C.2 DEFECT DETECTION

Defect detection is the task of classifying whether a code sample contains vulnerability points or not. We adopt the defect detection benchmark from CodeXGLUE (Lu et al., 2021) which curated data from the Devign dataset (Zhou et al., 2019). The dataset contains in total more than 27,000 annotated functions in C programming language. All samples are collected from popular open-source projects such as QEMU and FFmpeg. We follow Lu et al. (2021) and adopt 80%/10%/10% of the dataset as the training/validation/test split. For training, we set the learning rate as 2e-5, the batch size as 32, and the max sequence length as 512 to finetune the model for 10 epochs.

## C.3 CLONE DETECTION

The task of clone detection aims to detect whether any two code samples have the same functionality or semantics. We conduct experiments using the clone detection benchmark from CodeXGLUE (Lu et al., 2021). The benchmark is curated from the BigCloneBenchmark dataset (Svajlenko et al., 2014) and the resulting curated data consists of 901,724/416,328/416,328 examples for training/validation/test splits respectively. All samples are categorized into 10 different functionalities. To finetune CodeT5Mix on this task, we set the learning rate as 2e-5, the batch size as 10, and the max sequence length as 400. We use the Adam optimizer to finetune the model for 2 epochs.

## C.4 CODE COMPLETION

In code completion, a model is given a source sequence, containing a partial code sample, and is required to generate the remaining part of the code sample. We conduct experiments on line-level code completion, using two major benchmarks: PY150 (Raychev et al., 2016) and Github Java Corpus (Allamanis & Sutton, 2013).

PY150 (Raychev et al., 2016) consists of 150,000 Python source files collected from Github. Among these samples, Lu et al. (2021) selected 10,000 samples from different files from the test set of PY150 and then randomly sampled lines to be predicted for the code completion task. The average numbers of tokens in the source sequence and target sequence are 489.1 and 6.6 respectively.

Github Java Corpus (Allamanis & Sutton, 2013) contains over 14,000 Java projects collected from GitHub. Similarly to PY150, Lu et al. (2021) selected 3,000 samples from different files from the test set of the dataset and randomly sampled lines to be predicted for the code completion task. The average numbers of tokens in the source and target sequence are 350.6 and 10.5 respectively.

For both tasks, we set learning rate as 2e-5 and batch size as 32, and finetune the model for 30 epochs. We set the maximum sequence length of 1024 for the decoder. During inference, we employ beam search with a beam size of 5.

## C.5 CODE SUMMARIZATION

Code summarization is the task of generating a natural language summary of a code snippet. We use the task dataset from CodeXGLUE (Lu et al., 2021) which curated a code summarization benchmark from CSN data (Husain et al., 2019). The benchmark consists of six programming languages: Ruby, JavaScript, Go, Python, Java, and PHP. It is the same clean version of CSN data that we use for text-to-code retrieval tasks. For training, we set the maximum sequence length of source and target as 256 and 128, respectively. We use a learning rate of 2e-5, the batch size as 64 for 10 epochs of finetuning. We set the beam size as 5 in inference.

## C.6 CODE GENERATION

The task of code generation requires a model to generate a corresponding code snippet given a natural language description. We experiment on the Java ConCode benchmark (Iyer et al., 2018) which consists of 33,000 Java projects collected from Github. The benchmark has 100k/2k/2k samples in the training/validation/test split. Each code generation sample consists of a natural language description, code environment, and the corresponding code snippet. For training, we set the maximum sequence length of source and target as 320 and 200. We use a learning rate of 2e-5, the batch size as 32 for 30 epochs of finetuning. We set the beam size as 5 in inference.

Figure 5: Example of how to convert natural language solution into a Python program on GSM

| Problem | Problem | Problem |
|---|---|---|
| Toulouse has twice as many sheep as Charleston. Charleston has 4 times as many sheep as Seattle. How many sheep do Toulouse, Charleston, and Seattle have together if Seattle has 20 sheep? | Janet's ducks lay 16 eggs per day. She eats three for breakfast every morning and bakes muffins for her friends every day with four. She sells the remainder at the farmers' market daily for $2 per fresh duck egg. How much in dollars does she make every day at the farmers' market? | Eliza's rate per hour for the first 40 hours she works each week is $10. She also receives an overtime pay of 1.2 times her regular hourly rate. If Eliza worked for 45 hours this week, how much are her earnings for this week? |
| **Generated Program** | **Generated Program** | **Generated Program** |
| n0 = 4
n1 = 20
n2 = 2
t0 = n0 * n1
t1 = n2 * t0
answer = t1 + t0 + n1 | n0 = 3
n1 = 4
n2 = 16
n3 = 2
t0 = n0 + n1
t1 = n2 - t0
answer = t1 * n3 | n0 = 10
n1 = 40
n2 = 1.2
n3 = 45
t0 = n0 * n1
t1 = n0 * n2
t2 = n3 - n1
t3 = t2 * t1
answer = t0 + t3 |

Figure 6: Predictions of our model on GSM8K-Python

### C.7  MATH PROGRAMMING

Math Programming is the task of solving maths-based problems with programming. Compared to conventional code generation tasks such as ConCode (Iyer et al., 2018), this task focuses more on computational reasoning skills. The problem descriptions in this type of task are also more complex than conventional code generation tasks. We employ two major benchmarks for this tasks: MathQA-Python (Austin et al., 2021) and GradeSchool-Math (Cobbe et al., 2021).

MathQA-Python (Austin et al., 2021) is developed from the MathQA dataset (Amini et al., 2019) where given a mathematical problem description in natural language, a system is required to solve this problem using a domain-specific language. Austin et al. (2021) translated these programs into Python programs and filtered for cleaner problems. In total, MathQA-Python contains almost 24,000 problems, including 19,209/2,822/1,883 samples for training/validation/test splits.

GradeSchool-Math (Cobbe et al., 2021) (also known as GSM8K) has similar nature as MathQA. The benchmark focuses on problems with moderate difficulty that an average middle school student should be able to solve. In total, GSM data contains 8,500 problems, divided into 7,500 training and 1,000 testing problems. We translate the solution described in natural language to Python programs by following Austin et al. (2021) in the construction of MathQA-Python. Finally, we successfully converted 5,861 training samples. One conversion example can be found in Fig. 5. For qualitative results of CodeT5Mix, we provide our model predictions in Fig. 6 and Fig. 7.

For training, we set the maximum sequence length of source and target as 256 and 256 for MathQA-Python, and 246, 138 for GSM8k-Python. We use a learning rate of 2e-5, the batch size as 32 for 30 epochs of finetuning. During inference, we employ the beam size as 5 to get $pass@1$ results. For $pass@80$ and $pass@100$, we found they are quite sensitive to the diversity of the generation. We employ nucleus sampling with a temperature of 1.2 and top-$p$=0.95.

### D  RETRIEVAL-AUGMENTED CODE GENERATION

Developers often search for relevant code snippets from sources on the web such as GitHub or Stack-Overflow as references to aid their software development process. Motivated by this behaviour, we

| Problem | Problem | Problem |
|---|---|---|
| A shopkeeper sold an article offering a discount of 5% and earned a profit of 31.1%. What would have been the percentage of profit earned if no discount had been offered?
n0 = 5.0 n1 = 31.1 | What will be the difference between simple and compound interest at 14% per annum on a sum of rs. 1000 after 4 years?
n0 = 14.0 n1 = 1000.0 n2 = 4.0 | A full stationary oil tank that is a right circular cylinder has a radius of 100 feet and a height of 25 feet. Oil is pumped from the stationary tank to an oil truck that has a tank that is a right circular cylinder until the truck 's tank is completely filled. If the truck 's tank has a radius of 6 feet and a height of 10 feet, how far (in feet) did the oil level drop in the stationary tank?
n0 = 100.0 n1 = 25.0 n2 = 6.0 n3 = 10.0 |
| **Generated Program** | **Generated Program** | **Generated Program** |
| n0 = 5.0
n1 = 31.1
t0 = n1 + 100.0
t1 = 100.0 - n0
t2 = t0 * 100.0
t3 = t2 / t1
answer = t3 - 100.0 | n0 = 14.0
n1 = 1000.0
n2 = 4.0
t0 = n0 / 100.0
t1 = t0 + 1.0
t2 = n1 * t0
t3 = n2 * t2
t4 = t1**min(n2, 5)
t5 = n1 * t4
t6 = t5 - n1
answer = t6 - t3 | import math
n0 = 100.0
n1 = 25.0
n2 = 6.0
n3 = 10.0
t0 = math.pi * n0**2
t1 = math.pi * n2**2 * n3
answer = t1 / t0 |

Figure 7: Predictions of our model on MathQA-Python

explore a retrieval-augmented code generation setting, where given a natural language description, a retriever first retrieves similar candidates in a search codebase and then augments the input for the generator to produce the target code. Such retrieval-augmented generation (or retrieve-then-generate) paradigm has been widely used in open-domain question answering (Karpukhin et al., 2020) in NLP and recently extended to some code-related tasks such as code generation and summarization (Parvez et al., 2021) with significant improvements. As our CodeT5Mix is capable of both retrieval and generation, it can be seamlessly adapted as a unified retrieval-augmented generator. This can bring unique benefits such as less computational cost compared to prior work that employs a different retriever and generator. We evaluate CodeT5Mix on two Java and Python code generation datasets from the CodeXGLUE Lu et al. (2021) benchmark following Parvez et al. (2021).

Specifically, we leverage the bimodal encoder to encode the code snippet in the retrieval base and build a search index with the `faiss` library (Johnson et al., 2019). The search index is a set of representations (of 256 dimensions) for all the code snippets in the retrieval codebase. Let $(x_i, y_i)$ denote one training instance where $x_i$ is the input text description and $y_i$ is the corresponding target code snippet. we employ the same bimodal encoder to obtain the embedding of $x_i$ and retrieve top-$k$ similar code samples from the search base using the L-2 similarity metric, with $k$ being a hyperparameter. We ensure that the training example's target string ($y_i$) is not present in any of these $k$ retrieved samples.

After retrieving these top-$k$ relevant code samples, we combine them with a special token [SEP] and concatenate it to the end of the source input $x_i$. Unlike Parvez et al. (2021), we do not augment docstrings or text descriptions and only augment the code snippet for simplicity. We then finetune CodeT5Mix on this augmented dataset. During inference, we retrieve similar code samples from the search base and augment these to input $x_i$. For training, we set the maximum sequence length of source and target as 600 and 320. We use a learning rate of 2e-5, the batch size as 32 to finetune the model for 10 epochs. We set the beam size as 5 in inference.

We provide a case study in Fig. 8 for qualitative evaluation, where retrieval-augmented generator predicts the ground-truth function while both retriever and generator fail to do so. We further conduct an ablation study to analyze the effects of top-$k$ retrievals in Table 10. We found that increasing the number of retrievals can boost model performance which becomes saturated when $k$=5. This saturation is due to the maximum sequence length of 600, which might not be able to accommodate a large number of retrieved code samples.

## E  ADDITIONAL EXPERIMENTAL RESULTS

We provide the full results of code summarization in Table 11, and full ablation results of code summarization in Table 12.

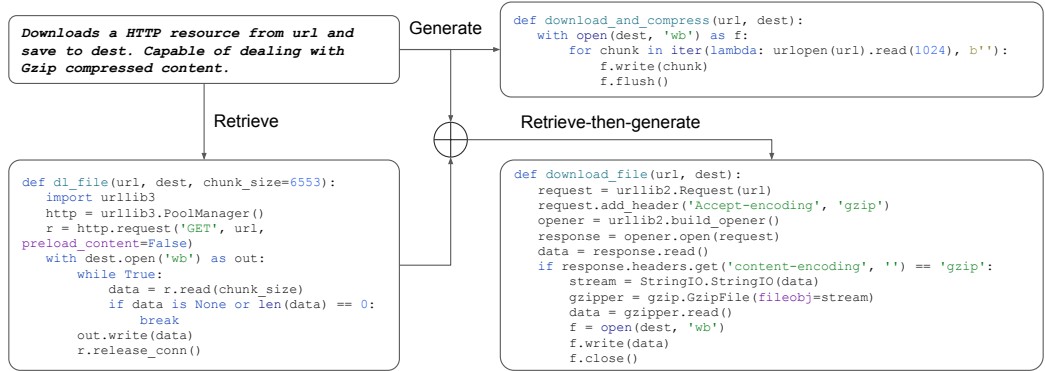

Figure 8: One example of retrieval-augmented code generation in Python dataset, where $\bigoplus$ denotes the concatenation operation. With the help of the top retrieved code, our CodeT5Mix-based retrieval-augmented generator produces the correct function.

Table 10: Effects of varying top-$k$ retrievals in retrieval-augmented code generation

| Model | Java | | | Python | | |
|---|---|---|---|---|---|---|
| | EM | B4 | CB | EM | B4 | CB |
| REDCODER-EXT (top-10) | 10.21 | 28.98 | 33.18 | 9.61 | 24.43 | 30.21 |
| CodeT5Mix-base (top-1) | 11.66 | **33.83** | 40.60 | 11.83 | 31.14 | 36.39 |
| top-2 | 11.57 | 33.26 | 40.74 | 11.78 | **31.21** | 36.58 |
| top-3 | 12.29 | 33.10 | 41.71 | 12.48 | 30.92 | 37.31 |
| top-4 | 12.42 | 32.08 | 41.94 | 12.73 | 30.40 | 37.60 |
| top-5 | **13.02** | 32.42 | **42.28** | **12.93** | 30.52 | **37.87** |
| top-10 | 12.86 | 31.38 | 42.24 | 12.84 | 29.79 | 37.79 |

Table 11: Full results (smoothed BLEU-4) on code summarization

| Model | Ruby | JavaScript | Go | Python | Java | PHP | Overall |
|---|---|---|---|---|---|---|---|
| RoBERTa | 11.17 | 11.90 | 17.72 | 18.14 | 16.47 | 24.02 | 16.57 |
| CodeBERT | 12.16 | 14.90 | 18.07 | 19.06 | 17.65 | 25.16 | 17.83 |
| PLBART | 14.11 | 15.56 | 18.91 | 19.30 | 18.45 | 23.58 | 18.32 |
| CoTexT | 14.02 | 14.96 | 18.86 | 19.73 | 19.06 | 24.58 | 18.54 |
| UniXcoder | 14.87 | 15.85 | 19.07 | 19.13 | 20.31 | 26.54 | 19.30 |
| CodeT5-base | 15.24 | 16.16 | 19.56 | 20.01 | 20.31 | 26.03 | 19.55 |
| CodeT5-large | 15.58 | 16.17 | 19.69 | 20.57 | 20.74 | 26.49 | 19.87 |
| CodeT5Mix-base | 15.51 | 16.27 | 19.60 | 20.16 | 20.53 | 26.78 | 19.81 |
| CodeT5Mix-large | 15.63 | 17.93 | 19.64 | 20.47 | 20.83 | 26.39 | **20.15** |

Table 12: Full ablation results (smoothed BLEU-4 scores) on code summarization

| Model | Ruby | JavaScript | Go | Python | Java | PHP | Overall |
|---|---|---|---|---|---|---|---|
| CodeT5Mix-base | 15.47 | 16.27 | 19.44 | 19.56 | 20.23 | 26.19 | 19.53 |
| no match decoder | 15.21 | 15.84 | 19.43 | 19.44 | 20.43 | 26.57 | 19.49 |
| no code/text decoder | - | - | - | - | - | - | - |
| separate decoder | 15.01 | 16.22 | 19.33 | 19.31 | 20.16 | 26.23 | 19.38 |
| shared decoder | 15.39 | 16.06 | 19.27 | 18.89 | 19.97 | 25.84 | 19.24 |

