# OpenReview forum: "CodeT5Mix: A Pretrained Mixture of Encoder-decoder Transformers for Code Understanding and Generation"
_ICLR.cc/2023/Conference — Submitted to ICLR 2023_

### Official Review · Reviewer_stVy · 2022-10-24

**Confidence:** 4
**Correctness:** 4
**Technical Novelty And Significance:** 2
**Empirical Novelty And Significance:** 2
**Recommendation:** 6

**Clarity, Quality, Novelty And Reproducibility:**

The paper is written in a clear and organized way. There are some technical details such as the equation for each loss readers can’t find in the paper. However, it’s not critical for reading.

**Strength And Weaknesses:**

Strength:
• The thorough evaluation shows that the proposed CodeT5Mix has strong performance against the CodeT5 baseline

Weakness:
• The contribution can be seem as an incremental extension to CodeT5 for enhancing the training with multi-task objectives. For the evaluation results, comparing to CodeT5 baseline, the gain is more incremental rather than break-through. Such a gain is largely expected when extending the original masked span prediction loss of CodeT5

**Summary Of The Paper:**

This paper proposes a multi-task training approach based on CodeT5 for solving multi-modality code related tasks. The model has multiple general and specialized components based on the task. The training contains two stages. In the first stage, the model trains on span denoising and CLM objectives. In the second stage, the model trains on contrastive learning, matching and generation objectives. Evaluation results show that the proposed model outperforms CodeT5 in similar conditions.

**Summary Of The Review:**

I lean towards an acceptation for this paper given the extensive study of model performance in experiments. However, on the significance and impact of the contribution, the modification to the pre-training objectives seem to be a natural extension (by training on multiple code related tasks) of the CodeT5 objective.

---

> ### Author Response · Authors · 2022-11-14
> **Response to Reviewer stVy**
>
> ### Q1: “The contribution can be seen as an incremental extension to CodeT5 for enhancing the training with multi-task objectives.”
> We want to emphasize that CodeT5Mix innovates from two perspectives compared to CodeT5: 1) a flexible mixture architecture that can operate in different modes including encoder-only, decoder-only, and encoder-decoder; 2) a mixture of self-supervised learning objectives to build a versatile model with better code understanding and generation capabilities. To make the overall approach effective in a large variety of code-related tasks, we further propose a weight-sharing design of task-specific experts to reconcile the interference of multi-task learning. Please refer to the “Response to the novelty concern” in the general response to all reviewers for more details.
> ### Q2: “For the evaluation results, comparing to CodeT5 baseline, the gain is more incremental rather than break-through.”
> We extensively evaluate our model on a wide range of code-related tasks, achieving new SoTA baselines in most cases. Particularly compared to CodeT5-base, our CodeT5Mix-base yields significant performance gains on many tasks, including 8 text-to-code retrieval tasks (+5.89 avg. MRR), 2 line-level code completion tasks (+6.45 EM on PY150 and +9.43 EM on JavaCorpus), and 2 math programming tasks (+14.10 Pass@80 on MathQA-Python and +12.10 Pass@100). Note that the performance of some tasks such as code clone detection has already become saturated and our results are still better than or on par with the SoTA performance. Please refer to the “Response to the model performance concern” in the general response to all reviewers.
> ### Q3: “The paper is written in a clear and organized way. There are some technical details such as the equation for each loss readers can’t find in the paper. However, it’s not critical for reading”
> Thanks for the comment! We provide detailed equations of the text-code contrastive learning and matching objectives in Appendix A.2.

---

> > ### Comment · Reviewer_stVy · 2022-11-16
> > **Response to the authors**
> >
> > Thanks for updating the loss in the appendix. I maintain my opinion leaning toward an acceptation for this paper.

---

### Official Review · Reviewer_3yBT · 2022-10-26

**Confidence:** 4
**Correctness:** 4
**Technical Novelty And Significance:** 3
**Empirical Novelty And Significance:** 3
**Recommendation:** 6

**Clarity, Quality, Novelty And Reproducibility:**

This paper is well-writtern and is of high quality. The proposed method is novel. And I expect that the results should be easy to reproduce.


**Strength And Weaknesses:**

Strength:
- The paper is well-written. The method is well explained with examples. Experiment results are extensive with analysis.
- The proposed method provides multiple encoder/decoders to solve different tasks in a mixture of expert manner, and could have the potential to avoid task interference.


Weakness:
- It is not clear to me that how many encoders and decoders there are in the proposed CodeT5Mix. Inferring from Section 3.1, there should be one encoder, three decoders (one for matching decoder, one for text generation decoder and one for code generation decoder). And 1st stage of pretraining is to pretrain one encoder-decoder, from 2nd stage, the decoders are then separated out. Is my understanding correct? Please provide clarifications on the architectures of the proposed CodeT5Mix.
- One simple baseline to CodeT5Mix is that, we can use three separate encoder-decoders initialized from 1st stage pretraining for subsequent stage of pretraining/finetuning tasks. Could the authors demonstrate the benefit of CodeT5Mix by comparing the results of this simple baseline?
- First stage of pretraining objectives seems the same as CodeT5. For later stages, how large gain can we get by using separate decoders compared with using only one decoder but with different sequence formatting for different tasks? I.e. are the separate decoders necessary? The paper also lacks this comparison.

**Summary Of The Paper:**

This paper proposes a mixture of encoder-decoder transformers, called CodeT5Mix, for code understanding and generation. It also proposes a number of pretraining tasks including denoising, causal LM, constrastive loss, matching to pretrain the CodeT5Mix. Extensive experiments on various tasks have been well performed.

**Summary Of The Review:**

Overall, I think this is a decent paper with novel method proposed and well-performed experiments. However, I have concerns on the proposed mixture of decoders, and would like to see more experiments to compare with simple baseline to justify the advantanage of mixture of decoders.

---

> ### Author Response · Authors · 2022-11-14
> **Response to Reviewer 3yBT**
>
> ### Q1: “It is not clear to me that how many encoders and decoders there are in the proposed CodeT5Mix.”
> In total, there is one bimodal encoder and three decoders (one for matching decoder, one for text generation decoder, and one for code generation decoder). In the first pretraining stage, a vanilla encoder-decoder Transformer model is trained using Github unimodal data. In the second stage, from the model checkpoint in the first stage, we initialize three decoders as described above. These decoders share intermediate attention layers and only have separate feed-forward layers that match corresponding learning objectives. From another perspective, we can view it as one decoder with three task-specific layers, which is illustrated in Figure 4 of the Appendix.
> ### Q2: “One simple baseline to CodeT5Mix is that, we can use three separate encoder-decoders initialized from 1st stage pretraining for subsequent stage of pretraining/finetuning tasks.”
> Yes, we indeed have such an ablation in Section 4.4. You can see a variant of “separate decoder” in Table 6, where we find that our base model consistently achieves performance gain than it, including +0.57 avg. MRR over 6 text-to-code retrieval tasks, +0.15 avg. BLEU-4 over 6 code summarization tasks, +0.4 CodeBLEU on one code generation task. This is probably due to the fact that fully separate decoders cannot enjoy the benefits of multi-task learning, while our weight-sharing scheme allows cross-task generalization from the joint training whereby the contextual representations in attention layers learned from one task can complement other tasks.
> ### Q3: “For later stages, how large gain can we get by using separate decoders compared with using only one decoder but with different sequence formatting for different tasks?”
> We indeed have such an ablation in Section 4.4 and you can see the comparison between “separate decoder” and “shared decoder” from the last two variants in Table 6. We can observe that these two variants have comparable performance and consistently lower performance than our base model in all 3 tasks, validating the effectiveness of our model design. Moreover, we find that the “separate decoder” only brings slight improvements in code summarization and generation tasks at a large parameter cost of two additional decoders, and no improvement on text-to-code retrieval tasks.

---

> ### Author Response · Authors · 2022-12-12
> **Friendly reminder to respond to author rebuttal**
>
> Dear Reviewer 3yBT,
>
> Thank you again for your review!  We are happy to hear that you appreciated CodeT5Mix's novelty, well-performed experiments, motivation, and presentation.
>
> Based on your thoughtful feedback, we wrote a detailed rebuttal covering the following points:
> * Clarification on the model architecture [Q1]
> * Clarification on ablation experiments [Q2,Q3]
>
> We would love to hear your thoughts about our rebuttal, including whether it sufficiently addresses your concerns and questions. If you believe that our rebuttal is satisfactory, it would be great if you could consider increasing your score. Any feedback is welcome and greatly appreciated!
>
> Sincerely,
>
> Paper5717 Authors

---

### Official Review · Reviewer_ZFgG · 2022-10-26

**Confidence:** 4
**Correctness:** 3
**Technical Novelty And Significance:** 2
**Empirical Novelty And Significance:** 3
**Recommendation:** 5

**Clarity, Quality, Novelty And Reproducibility:**

The paper is largely well-written, though some technical details are somewhat obfuscated. In particular:
* Sect 3.2.1 and 3.2.2 should state more clearly what encoder/decoder outputs are.
* Sect. 3.2.3 could profit from more clarity on parameter sharing arrangements: it starts by stating that the decoders share model parameters, but then walks this back. It's unclear what the weight-sharing scheme on the encoder side is, but from Fig. 3, it seems that all parameters are shared. The paper would be strengthened by an explanation on how the final scheme was decided on.

The experiment design seems largely solid, with three notes:
* The github-code data has substantial duplication issues (see https://twitter.com/miltos1/status/1497126435261083649). Have you done any work to mitigate these?
* The experimental results should state parameter counts for all models, to make it easier to disentangle size from performance.
* No error bars are reported, which is problematic given that differences between model results are often very, very small. This is especially noteworthy for the ablation results in Table 6.

Overall, the proposed pretraining multi-task method is not particularly novel, but the idea of performing limited parameter sharing for this in the code space is.

Notes:
* page 3: "Additionally, encoder-decoder models are not ideal for autoregressive tasks like code completion (Lu et al., 2021), in which existing SoTA methods are mostly based on decoder-only models": this may be an artefact of how code completion models are benchmarked (usually with context only "to the left"). Papers such as https://arxiv.org/pdf/2204.05999.pdf indicate that there are substantial differences in this setting, and encoder-decoder models support this case well. Industry reports such as https://ai.googleblog.com/2022/07/ml-enhanced-code-completion-improves.html also indicate that encoder-decoder models may be more suitable for the code completion setting.
* Fig. 2 and 3 contain information that is not accessible to the colourblind (or reviewers working on greyscale copies of the paper). Please fix this this.
* Table 2: Any reason why the $95.2$ for UniXcoder is not bolded?

**Strength And Weaknesses:**

* (+) Substantial experiments covering a wide variety of different tasks
* (~) Improvements over simpler baselines are quite modest
* (-) No experiments validating core architectural choices (e.g., using task-specific expert layers vs. using same number of parameters in fully shared model)

**Summary Of The Paper:**

A pretraining strategy for models performing different tasks on code is presented. It follows the idea of implementing multiple related architectures using partial weight-sharing. In particular, the core architecture is an encoder-decoder Transformer model. Different tasks share all model parameters apart from the feed-forward layers in the Transformer (i.e., the weights for computing attention scores are shared). Pretraining proceeds by first considering causal language modeling and span denoising tasks. In a second stage, it employs the code/text pairs from the CodeSearchNet dataset to train on four related tasks (contrastive learning, matching, text->code and code->text). Finally, the model is fine-tuned on different downstream tasks.
In an evaluation, the model shows modest improvements over baselines on a range of tasks from the literature. An additional ablation study is performed to analyze the effect of different choices in pretraining/parameter sharing.

**Summary Of The Review:**

A reasonably-well executed paper with minor problems in clarity. In practice, I have doubts that the proposed method is going to be adopted by a wider set of researchers, given the complexities of managing the proposed parameter-sharing scheme and the modest gains. I'd tend to accept this because there's nothing _wrong_ with the paper, but I would also not champion it as an important contribution that needs to be presented at ICLR.

Update after discussion period: while my overall opinion hasn't shifted substantially, I'm now leaning towards rejecting the paper, given that the validity of the empirical results remains somewhat in doubt given the lack of deduplication of the data and the numerous small concerns I and the other reviewers voiced on the presented experiments (primarily the lack of error bars, but also a lack of deeper investigation of the architecture and objective choices).

---

> ### Author Response · Authors · 2022-11-14
> **Response to Reviewer ZFgG (1/n)**
>
> ### Q1: “ No experiments validating core architectural choices (e.g., using task-specific expert layers vs. using same number of parameters in fully shared model)”
>  We have conducted a comprehensive ablation study in 3 representative tasks and 13 sub-datasets to validate the core architectural design and weight-sharing strategy (see Table 6). We improve the presentation and explanation in the revised paper, where we can find the “shared decoder” variant (the last row in Table 6), which is a model with fully shared decoders. In most tasks, we find our full CodeT5Mix model with task-specific expert layers consistently achieves performance gain than this variant, i.e., +0.55 avg. MRR over 6 text-to-code retrieval tasks, +0.29 avg. BLEU-4 over 6 code summarization tasks, +0.59 CodeBLEU on code generation. Note that although CodeT5Mix incurs additional insignificant parameter costs by introducing multiple expert layers, such extra cost can be waived when applied to a specific downstream task as we only need to activate a single task-specific expert.
> ### Q2: “Improvements over simpler baselines are quite modest”
> We respectfully disagree.  We extensively evaluate our model on a wide range of code-related tasks and show that CodeT5Mix achieves significant improvements over prior SoTA baselines in most cases, including 8 text-to-code retrieval tasks, 2 line-level code completion tasks, 2 math programming tasks, and 2 retrieval-augmented code generation tasks. Note that the performance of some tasks such as code clone detection has already become saturated and our results are still better than or on par with the SoTA performance. Please refer to the “Response to the model performance concern” in the general response to all reviewers.
> ### Q3: “The github-code data has substantial duplication issues”
> Thanks for your comment! We want to highlight that we filtered the Github pretraining data and removed any files that originate from the same GitHub repositories as the testing data (e.g. CodeSearchNet test splits). Due to the space constraints, we move this information from Section 4.1 in the original version to Appendix B.1 in the revised paper.
> ### Q4: “The experimental results should state parameter counts for all models”
> Thank you for your feedback! We want to highlight the model sizes of major baseline models below. We also added more details of the model sizes in the revised paper in Section 4:
> - CodeBERT, GraphCodeBERT, SYNCOBERT, UniXcoder are based on RoBERTa-base and have 125M parameters
> - GPT-2 and CodeGPT have 124M and PLBART has 140M
> - CodeT5 and CodeT5Mix models are evaluated at 2 sizes: base (220M) and large (770M). Note that for CodeT5Mix, it only uses roughly half of the parameters when acting in encoder-only or decoder-only mode.
>
> We also compare our models with very large-scale language models for Math Programming tasks. Specifically, in Table 5, we added the model parameters for all models, such as Codex, LaMDA, PaLM-coder, etc. whose model sizes range from 6B to 540B.
> ### 5: “No error bars are reported”
> Due to the high computation cost of large pretrained models, we performed one pretraining run following the common practice such as UniXcoder. In our preliminary experiments, we found the pretraining to be insensitive to random seeds. From the full ablation results in Table 6, we demonstrate that the performance gains are indeed more significant in many cases. Moreover, the consistent performance gap from a broad set of datasets (6 text-to-code retrieval datasets, 6 code summarization datasets, and 1 code generation dataset) should compensate for the randomness and thus well validate the effectiveness of our overall model design.

---

> > ### Author Response · Authors · 2022-11-14
> > **Response to Reviewer ZFgG (2/n)**
> >
> > ### Q6: “Papers… indicate that there are substantial differences in this setting, and encoder-decoder models support this case well. Industry reports … also indicate that encoder-decoder models may be more suitable for the code completion setting.”
> > Thank you for suggesting these related papers! However, a closer look at [1] shows that the foundation models being used in this work actually follow decoder-only Transformer architecture. Their approach relies on causal LM pretraining tasks and focuses on generative downstream tasks only (e.g. infilling tasks). In [2], the article focuses on engineering efforts to use an encoder-decoder LM for code completion. This work only demonstrates the performance when the model is enhanced with a semantic engine, without comparison with a decoder-only model. In our work, our focus is to develop a unified encoder-decoder model that can achieve SoTA results in both generative and understanding downstream tasks. We added these comparisons to these two papers and explained more in the revised paper in the Related Work section.
> >
> > [1] InCoder: A Generative Model for Code Infilling and Synthesis https://arxiv.org/pdf/2204.05999.pdf
> >
> > [2] ML-Enhanced Code Completion Improves Developer Productivity https://ai.googleblog.com/2022/07/ml-enhanced-code-completion-improves.html
> >
> > ### Q7: Practical concern of “the complexities of managing the proposed parameter-sharing scheme”.
> > We provide another view of our mixture architecture in Figure 4 in Appendix, where we can observe that it can be interpreted as a unified encoder-decoder with task-specific FFN experts in the decoder. When finetuning on downstream tasks in the encoder-decoder mode, only one of the task-specific experts is activated, thereby incurring zero extra parameter cost compared to a standard encoder-decoder model.
> > ### Others:
> > #### “Fig. 2 and 3 contain information that is not accessible to the colourblind (or reviewers working on greyscale copies of the paper). Please fix this.”
> > Thanks for pointing out this issue! We added more explanations in the caption of Figure 3 in the revised paper to help readers understand the weight sharing strategy without relying on the color annotation.
> > #### “The paper is largely well-written, though some technical details are somewhat obfuscated.”
> > Thank you for your comments! We have revised the paper and added more clarification in the corresponding sections. Particularly, we add further model explanations in Appendix A with another model illustration and detailed formulas for some pretraining objectives.
> > #### “Table 2:... for UniXcoder is not bolded?”
> > Thanks for your feedback! We fixed this issue in the revised paper.

---

> > > ### Comment · Reviewer_ZFgG · 2022-11-15
> > > **Paper Discussion**
> > >
> > > Re your answer to Q6: My claim was not that InCoder is an Encoder/Decoder model, but that the results in Table 1 in the paper indicate that left-to-right ("LM-style") decoding is not doing well in the realistic setting of code completion with context after the cursor. Again, my point is that your claim "encoder-decoder models are not ideal for auto-regressive generation tasks like code completion" is not well-founded, and that results that you are reporting may have more to do with the fact that many experiments are on the unrealistic setting in which there is no context after the completed code.

---

> > > > ### Author Response · Authors · 2022-11-16
> > > > **Response to Paper Discussion**
> > > >
> > > > Thank you for the feedback. We will make our claim preciser to focus on the auto-regressive code completion (different from the infilling-based code completion). We follow the UniXcoder work to include the next-line code completion task from the CodeXGLUE benchmark, aiming to evaluate our model’s auto-regressive generation capability instead of the infilling capability like Incoder.
> > > > We do recognize that there are two settings of code completion (whether the after context is used) but do not agree that only the infilling-based code completion is realistic and auto-regressive code completion is unrealistic. For the latter setting, it has been widely studied in both academic and industry domain [1], and many recent code completion tools are based on GPT-style LMs such as Codex and CodeGen, which are trained and employed in an auto-regressive manner without using the after context.
> > > >
> > > > [1] IntelliCode Compose: Code Generation using Transformer. ESEC/FSE 2020.

---

> > ### Comment · Reviewer_ZFgG · 2022-11-15
> > **Paper Discussion**
> >
> > Thank you for your extensive reply.
> >
> > Re your answer to Q1: I think a fundamental issue here is that you are claiming that using roughly twice as many parameters is "insignificant". What I assume you mean is that the _computational_ cost is not much higher, given that you are not using all of the additional parameters in all situations. However, the paper does not make this claim - it specifically talks about "keeping the model parameters affordable". Do you want to rephrase this?
> > At the moment, your experiments in table 6 do not show what I asked for: the parameter counts for these models are wildly different.
> >
> > Re your answer to Q2: Your answer on this does not reflect different parameter counts or computational budgets. The contribution you are proposing is a weight-sharing scheme paired with a pretraining strategy. To show that this yields improvements that you couldn't achieve by increasing parameter counts alone, you _cannot cite the results of CodeT5Mix-large_: that model is by far larger than the baselines you are comparing to. I understand that comparing encoder/decoder models, encoder-only, and decoder-only models is hard, especially when throwing in the complication of expert layers. However, a simple way out would be to compare computational effort, i.e., counting FLOPs for all models. Until you do that, a reasonable read of your experimental data is "CodeT5Mix-base is slightly bigger than UniXcoder, and does slightly better". [All of this still neglects the influence of training length for the different models vs. differences in objectives, see the recent Chinchilla paper...]
> >
> > Re your answer to Q3: Can you please confirm that you have not removed any duplicates, apart from removing overlap with CodeSearchNet?

---

> > > ### Author Response · Authors · 2022-11-16
> > > **Response to Paper Discussion**
> > >
> > > Thank you for taking the time to review our paper and for your quick response!
> > >
> > > ### 1: Clarification on parameter counts in Table 6 and our weight sharing strategy.
> > > We do NOT claim that "twice as many parameters is insignificant".  Compared to a standard encoder-decoder model, our CodeT5Mix introduces only two FFN experts in the decoder (see Figure 4 in the revised paper for better illustration), which did not cost twice as many parameters. Take the CodeT5Mix-base in Table 6 as an example, each FFN expert introduces an extra 28M parameter compared to CodeT5-base (220M), which is the insignificant overhead we refer to.
> > >
> > > Specifically, for a standard encoder-decoder model (i.e. CodeT5-base) , it has 198M parameter (apart from the shared embedding), where the encoder has 85M parameter and the decoder with a FFN expert (28M) has 113M parameter.
> > > Below we list the overall parameter counts (#P) and those which are activated and really used in each task for all variants in Table 6 (note that we exclude the embedding parameter as all variants include the same counts):
> > >
> > >
> > > | Model                | Overall #P     | Text-to-code Retrieval (activated #P) | Code Summarization (activated #P) | Code Generation (activated #P) |
> > > | -------------------- | -------------- | ------------------------------------- | ---------------------------- | ------------------------- |
> > > | CodeT5Mix-base       | 198M + 2\*28M  | 198M                                  | 198M                         | 198M                      |
> > > | No match decoder     | 198M + 28M     | 85M (only encoder is used)            | 198M                         | 198M                      |
> > > | No code/text decoder | 198M           | 198M                                  | 198M                         | 198M                      |
> > > | Separate decoder     | 198M + 2\*113M | 198M                                  | 198M                         | 198M                      |
> > > | Shared decoder       | 198M           | 198M                                  | 198M                         | 198M                      |
> > >
> > >
> > > From the above table, we can observe that compared to the “shared decoder” variant, MixCodeT5-base introduces 2*28M additional overall parameters and activates the same number of parameters (i.e. same computational cost) in each downstream task.
> > >
> > > ### 2: Performance improvement with parameter counts.
> > > First of all, we want to emphasize that it is difficult (or impractical in our case) to pretrain many model variants to compare baseline models with different architectures using the same parameter counts in a fully fair way. Instead, we build models at the similar size as CodeT5 models and ensure the same number of parameters activated in downstream tasks (thereby with the same computational cost).
> > >
> > > Secondly, we argue that increasing parameter counts alone cannot guarantee performance improvement. For instance, CodeT5-base achieves a much lower performance than UniXcoder on 8 text-to-code retrieval tasks (Table 1) and 2 next-line code completion tasks (Table 3) with a larger model size (220M vs. 125M), and our CodeT5Mix models outperform many much larger models such as LaMDA 137B on 2 math programming tasks.
> > >
> > > Thirdly, from a reasonably fair comparison between CodeT5Mix-base and its CodeT5-base counterpart (at the similar scale and with the same computational cost), we can find that CodeT5Mix-base achieves significant performance gains in many cases, including 8 text-to-code retrieval tasks (+5.89 avg. MRR), 2 line-level code completion tasks (+6.45 EM on PY150 and +9.43 EM on JavaCorpus), and 2 math programming tasks (+14.10 Pass@80 on MathQA-Python and +12.10 Pass@100 on GSM8K-Python).
> > >
> > > Last but not least, we want to highlight our contribution to build a larger pretrained LM for code to further push the performance limits of many code-related tasks. We will make the pretrained model checkpoints publicly available to benefit future research.
> > >
> > > ### 3: Clarification on data deduplication.
> > >
> > > As stated in Appendix B.1, we did filter out the overlapped subsets with CodeSearchNet and other downstream tasks by checking their GitHub repositories. One exception is the math programming tasks where they do not originate from GitHub.
> > >
> > > For potential remaining duplicates within the pretraining GitHub data, we did not perform further deduplication and directly employed [this dataset](https://huggingface.co/datasets/codeparrot/github-code) in which duplicates are filtered out based on exact match ignoring the whitespaces. However, we do not expect any remaining duplication will impact our model performance significantly. We made this issue aware in our paper (see Appendix B.2).
> > >
> > > Please let us know if you have any additional questions or concerns.

---

> ### Author Response · Authors · 2022-12-12
> **Friendly reminder to respond to author follow-up rebuttal**
>
> Dear Reviewer ZFgG,
>
> Thank you again for your review!  Based on your constructive feedback, we wrote a detailed follow-up rebuttal covering the following points:
> * Clarification on parameter counts in Table 6 and our weight sharing strategy
> * Clarification of performance improvement with different parameter counts
> * Clarification on data deduplication
> * Discusion on two settings of code completion
>
> We would love to hear your thoughts about our rebuttal, including whether it sufficiently addresses your concerns and questions. If you believe that our rebuttal is satisfactory, it would be great if you could consider increasing your score. Any feedback is welcome and greatly appreciated!
>
> Sincerely,
>
> Paper5717 Authors

---

### Official Review · Reviewer_EgG1 · 2022-10-27

**Confidence:** 4
**Correctness:** 2
**Technical Novelty And Significance:** 2
**Empirical Novelty And Significance:** 1
**Recommendation:** 3

**Clarity, Quality, Novelty And Reproducibility:**

Clarity:

Key details about the baseline models and training data seem to be missing

Quality:

Error bars are not given

Novelty:

The text-code contrastive learning/matching objective is marginally novel but it's not clear whether any significant gains are realized from adding it. Furthermore I can't find the motivation for having the matching objective given that the contrastive learning objective is already there.

**Strength And Weaknesses:**

Strengths:

The idea of exploring multiple pretraining objectives as well as weight sharing for different tasks is interesting. The evaluation is done on multiple tasks which is good.

Weaknesses:

(1) Causal language modeling and masked language modeling have been shown to be a very strong baselines for code generation and understanding tasks. It's unclear whether (3), (4), (5) add much. Indeed the ablation they do indicates that at least (4), (5) as well as weight sharing do not make a big performance difference. Thus we don't have good evidence of performance improvement from the more novel pretraining objectives.

(2) No error bars are given making it hard to assess the significance of improvements, especially for ablation experiments where the difference is minor

(3) Perhaps I missed it but I do not see the model sizes for the baseline models mentioned. Do they have the same number of parameters as the CodeT5Mix models? If not, then a smaller parameter count for those models would also account for the performance difference given the known language model scaling laws.

(4) What was the training data for the baseline models?

(5) Again perhaps something I missed, but I do not see any mention of the training data being deduped against the test data. Was that done?

(6) It is not clear to me why the text matching objective is necessary given the contrastive objective? In both cases you are trying to make the embeddings of the matching pairs closer and push the non-matching pairs further. Motivation for this should be provided.

**Summary Of The Paper:**

The paper uses a mixture of encoder-decoder transformers for code understanding and generation tasks. It in addition uses multiple pretraining objectives to train the mixture. The following pretraining objectives are standard and have been explored in previous works -- in particular (1) span denoising, (2) causal language . They also have (3) contrastive learning though which they apply on text-code pairs which is novel. They have a (4) text matching objective where they try and predict whether a particular text-code pair is a positive or negative. They also have the pretraining task of (5) predicting code from text and vice versa. For task specific decoders, they do weight sharing

They show good improvements for multiple baselines across multiple tasks. They also do an ablation study where they individually remove (4) and (5) and show minor drops in performance. They also have an ablation with and without weight sharing.

**Summary Of The Review:**

The ideas have some marginal novelty but the ablations are not thorough and seem to indicate that the novel objectives at best have very minor benefits. Motivation for the matching objective is not provided and error bars are not given. It's also not clear what the model sizes of the baseline models as well as training data for them was. It is also unclear whether the test data was present or not in the training data.

---

> ### Author Response · Authors · 2022-11-14
> **Response to Reviewer EgG1 (1/n)**
>
> ### Q1: “ The text-code contrastive learning/matching objective is marginally novel but it's not clear whether any significant gains are realized from adding it.”
>
> First of all, we want to emphasize that our novelty is not in each individual pretraining objective, but a unified pretraining framework to cohesively integrate multiple pretraining tasks. Our CodeT5Mix innovates from both model architecture and learning objective perspectives. Please refer to the “Response to the novelty concern” in the general response to all reviewers for more details.
>
> For the benefit of adding text-code contrastive learning, this can be interpreted from the large performance gap between CodeT5 and CodeT5Mix on text-to-code retrieval tasks in Table 1, where we observe that CodeT5Mix-base yields +5.31 avg. MRR over 8 text-to-code retrieval tasks than CodeT5-base. We empirically found that such performance gain is mainly attributed to the text-code contrastive learning. We will explicitly add this ablation in the final revision.
>
> For the benefit of adding text-code matching, this can be observed from the ablation study in Section 4.4 (see the updated paper with more detailed results and explanations). From Table 6, we can see that the matching decoder is critical to the text-to-code retrieval tasks, which is demonstrated by a large performance drop of a 1.65 absolute average MRR score over 6 datasets. For other ablations, we also observe consistent performance drops in all 3 different tasks across 13 datasets in total (as code summarization also has 6 datasets), which confirms the effectiveness of our overall model design.
>
> ### Q2: “No error bars are given making it hard to assess the significance of improvements”
>
> From the full ablation results in Table 6, we demonstrate that the performance gains are indeed more significant in many cases. Moreover, the consistent performance gap from a broad set of datasets (6 text-to-code retrieval datasets, 6 code summarization datasets, and 1 code generation dataset) should compensate for the randomness and thus well validate the effectiveness of our overall model design. Due to the high computation cost of large pretrained models, we performed one pretraining run following the common practice such as UniXcoder. In our preliminary experiments, we found the pretraining to be insensitive to random seeds.
>
> ### Q3: “I do not see the model sizes for the baseline models mentioned”
> Thank you for your feedback! We want to highlight the model sizes of major baseline models below. We also added more details of the model sizes in the revised paper in Section 4:
> - CodeBERT, GraphCodeBERT, SYNCOBERT, UniXcoder are based on RoBERTa-base and have 125M parameters
> - GPT-2 and CodeGPT have 124M and PLBART has 140M
> - CodeT5 and CodeT5Mix models are evaluated at 2 sizes: base (220M) and large (770M). Note that for CodeT5Mix, it only uses roughly half of the parameters when acting in encoder-only or decoder-only mode.
>
> We also compare our models with very large-scale language models for Math Programming tasks. Specifically, in Table 5, we added the model parameters for all models, such as Codex, LaMDA, PaLM-coder, etc. whose model sizes range from 6B to 540B.
>
> ### Q4: “What was the training data for the baseline models?”
>
> Due to the space constraints, we did not describe the details of training data for the baseline models, as different baselines might have different pretraining data, e.g. UniXcoder employs CodeSearchNet in six PLs (one common data used by many models) while PLBART employs much large-scale Java and Python data from GitHub BigQuery. Please refer to the original papers of the baselines for more details.
> ### Q5: “I do not see any mention of the training data being deduped against the test data”
> Thanks for your comment! We want to highlight that we filtered the Github pretraining data and removed any files that originate from the same GitHub repositories as the testing data (e.g. CodeSearchNet test splits). Due to the space constraints, we move this information from Section 4.1 in the original version to Appendix A in the revised paper.

---

> > ### Author Response · Authors · 2022-11-14
> > **Response to Reviewer EgG1 (2/n)**
> >
> > ### Q6: “Motivation for the matching objective is not provided…why the text matching objective is necessary given the contrastive objective?”
> > As described in Section 3.2.2 in the original paper, the matching task “aims to learn text-code bimodal representations that capture the fine-grained alignment between text and code modalities”.  While the contrastive objective can improve the representations of code/text, it is not ideal to learn a bimodal representation for a text-code pair. Therefore, we introduce a matching decoder to capture the fine-grained cross-modal alignments between text and code through its cross-attention layers. In the vision-language domain, the combination of contrastive learning and matching losses has been shown to be effective to improve text-image retrieval performance by [1].
> >
> > From Table 6, we can observe that the matching decoder is indeed particularly beneficial to the downstream text-to-code retrieval tasks, reflected by an average of 1.65 absolute MRR score over 6 datasets dropped after removing it. Please refer to our response to R1-eBFF in Q5 for more explanations on the justification of the text-code matching decoder. We also elaborate more on its motivation in the Introduction section of the revised paper.
> >
> > [1] Align before Fuse: Vision and Language Representation Learning withMomentum Distillation. (NeurIPS ‘21)

---

> ### Author Response · Authors · 2022-12-12
> **Friendly reminder to respond to author rebuttal**
>
> Dear Reviewer EgG1,
>
> Thank you again for your review!  Based on your constructive feedback, we wrote a detailed rebuttal covering the following points:
> * Novelty of CodeT5Mix and the benefits of adding text-code contrastive learning and matching objectives [Q1]
> * Clarity items including error bars, parameter counts and training data for baseline models, and data deduplication [Q2-Q5]
> * Motivation of the matching objective [Q6]
>
> We would love to hear your thoughts about our rebuttal, including whether it sufficiently addresses your concerns and questions. If you believe that our rebuttal is satisfactory, it would be great if you could consider increasing your score. Any feedback is welcome and greatly appreciated!
>
> Sincerely,
>
> Paper5717 Authors

---

### Official Review · Reviewer_eBFF · 2022-11-03

**Confidence:** 2
**Correctness:** 3
**Technical Novelty And Significance:** 3
**Empirical Novelty And Significance:** 3
**Recommendation:** 6

**Clarity, Quality, Novelty And Reproducibility:**

**Clarity**

There are few questions that needs to be clearly answered in the paper.
1. What is the justification for text-code matching decoder? Can't you do that task using the [CLS] token representation from encoder?
2. Why two stage training? What happens when you combine all the objectives in on stage (say, stage 2)? Specially, since two CLM objectives are also in 2nd stage? If you really insist on two stage training, what is the point of using those two CLM is 2nd stage, those tasks are already sufficiently trained in first stage?
3. How are the losses combined in the second stage?
4. What are different baselines for the ablation studies?

**Quality and Novelty**
1.  **+** Such an ensemble type model for code is novel.
2.  **+** Main concepts of the paper are well written and is an easy read.
3.  **-** Empirical results are not very well explained to show how CodeT5-mix is solving the problem the authors stated in their introductory arguments.

**Reproducibility**
While the authors provide some detail of the pretraining and finetuning, the code and data are not made public. (the authors promised to do so, upon acceptance, perhaps)


**Strength And Weaknesses:**

**Strengths**
1. The idea of **Ensemble of different models** and sharing the weights between them is interesting and innovative. Each of these models are equipped to deal with a specific information source/task.
2. Very thorough evaluation on downstream tasks.
3. Impressive results.
4. Fairly well written paper.
5. Trained two different sized models.

**Weakness**
1. While I agree with the motivation of the paper, there are few things that are confusing. First, the authors argue that Encoder-Decoder models are suboptimally equipped to do autoregressive generation tasks. That argument depends on what is the pretraining objective of such models. For instance, I agree that CodeT5 does not teach to generate the whole code autoregressively, but PLBART, SPT-code [1], NatGen [2] they are equipped with autoregressive generation capacity. The authors should compare with these models. Moreover, I am not understanding what part of CodeT5-mix is overcoming the problem the authors argued that PLBART, SPT-code, NatGen cannot. I don't just mean comparing the results, but also comparing on a conceptual level.
2. I commend the authors on the impressive results throughout the paper. However, the retrieval augmented generation part looks a bit disconnected from the paper. I believe it could be better connected to the paper since it shows off the capacity of both code understanding (retrieval) and generation. Also, Parvez et al. showed evaluation on Retrieval Augmented Code Summarization ([table 4 from here](https://arxiv.org/pdf/2108.11601.pdf)). Why not show such evaluation using CodeT5-mix. Since the input of summarization is code, this evaluation will really show CodeT5-mix's understanding capability.
3. It is not clear whether we need multiple decoders. What would happen if you trained a single encoder-decoder with all the training objectives you have on the same pretraining dataset you have? I think you have shown something very close in the ablation, but the options in ablation are not clear. In light with that, I request the authors clearer explanation on the ablation.


**References**

[1] Changan Niu, Chuanyi Li, Vincent Ng, Jidong Ge, Liguo Huang, and Bin Luo. 2022. [SPT-code: sequence-to-sequence pre-training for learning source code representations](https://arxiv.org/abs/2201.01549). In Proceedings of the 44th International Conference on Software Engineering (ICSE '22).

[2]  Saikat Chakraborty, Toufique Ahmed, Yangruibo Ding, Premkumar Devanbu, and Baishakhi Ray. 2022. [NatGen: Generative pre-training by" Naturalizing" source code](https://arxiv.org/abs/2206.07585). In Proceedings of the ACM Joint Meeting on European Software Engineering Conference and Symposium on the Foundations of Software Engineering (ESEC/FSE' 22).


**Summary Of The Paper:**

This paper proposed CodeT5-Mix - a pretrained model consisting of mixture of encoders and decoders for code understanding and generation tasks. The authors argue that encoder-decoder based pretrained models such as PLBART, CodeT5 performs suboptimally in different tasks. To overcome such problem, the authors proposed mixing and matching different encoders and decoder for different information modalities (i.e., code, text) and flexibly combine them based on the specifics of the finetuning tasks. The ensemble of the encoder-decoders is trained jointly based on several training objectives. Some of these modules (the authors talked about the decoders) share weights. The authors performed thorough evaluation on a large number of downstream tasks and showed SOTA performances.

**Summary Of The Review:**

While the idea of the paper is innovative and the results are impressive, there are some flaws in the explanation and scholarly arguments (please see the details in the evaluation above).

---

> ### Author Response · Authors · 2022-11-14
> **Response to Reviewer-eBFF (1/n)**
>
> ### Q1: “I agree that CodeT5 does not teach to generate the whole code autoregressively, but PLBART, SPT-code [1], NatGen [2] they are equipped with autoregressive generation capacity.”
> Thank you for your comment! We observe that prior work such as PLBART, SPT-code [1], and NatGen [2] are equipped with autoregressive generation capacity. However, our approach is different from these approaches in two major aspects:
> - Learning objectives: PLBART, SPT-Code [1], and NatGen [2] are trained mostly with variants of denoising objectives. While these objectives can pretrain the models to denoise spans of tokens, they are designed to recover spans of limited lengths given the surrounding contexts. However, they are not ideal for pretraining models to generate long sequences such as complete program generation for Math Programming tasks. In CodeT5Mix, we propose to combine denoising objectives with 2 variants of causal language modeling and require models to generate longer output sequences (either from scratch or conditioned on randomly sampled prior context).
> - Pretraining process: while prior work is limited to a single pretraining stage, we propose a two-stage approach to leverage both unimodal and bimodal data for pretraining. The unimodal data is publicly available in massive amounts and is used to bootstrap the models to understand code semantics. The resulting model checkpoint continues to be pre-trained with bimodal data, including parallel train-text pairs. This stage-wise pretraining can efficiently pretrain models with multimodal data in two stages, each using specific learning tasks to support code generation/understanding downstream tasks.
>
> We also explain these points and cited the papers in the Related Work section in our revised paper.
>
> [1] SPT-code: sequence-to-sequence pretraining for learning source code representations. In Proceedings of the 44th International Conference on Software Engineering (ICSE '22).
>
> [2] NatGen: Generative pretraining by" Naturalizing" source code. In Proceedings of the ACM Joint Meeting on European Software Engineering Conference and Symposium on the Foundations of Software Engineering (ESEC/FSE' 22).
>
> ###  Q2: “the retrieval augmented generation part looks a bit disconnected from the paper”
>
> We include the evaluation of retrieval-augmented code generation to demonstrate that CodeT5Mix can be seamlessly employed as a unified semi-parametric retrieval-augmented generator as it can do both code retrieval and generation. Notably, the retrieval-augmented generation (or retrieve-then-generate) paradigm has been widely used in open-domain question answering [3] in NLP and recently extended to some code-related tasks such as code generation and summarization [4] with significant improvements.
>
> We evaluate CodeT5Mix-base on the more challenging code generation tasks following [4] in
> Section 4.5. From Table 7, we showed that within a unified model, CodeT5Mix outperforms a SoTA baseline REDCODER-EXT in [4] by a large margin, which requires two different models of a code retriever and generator. Besides, we observe that the retrieval-augmented generative results are significantly better than either retrieval-based or generative results. We believe our CodeT5Mix-based retrieval augmented generation is an important and general approach that can be transferred to any programming language and many other tasks.
>
> For qualitative evaluation, we further provide a case study in Figure 8 in the Appendix to better illustrate how retrieval-augmented generation helps the code generation compared to either retrieval-based or generative models.
>
> [3] Dense passage retrieval for open-domain question answering. In Proceedings of the 2020 Conference on Empirical Methods in Natural Language Processing (EMNLP ‘2020)
>
> [4] Retrieval augmented code generation and summarization. (EMNLP ‘2021)
>
> ### Q3: “Why not show such evaluation (retrieval-augmented code summarization in REDCODER) using CodeT5Mix”
> Thanks for the suggestion! Actually we did some preliminary retrieval-augmented generation experiments on code summarization before. We observed it also yields better results than retrieval-based and generative results, but with a slighter performance gain compared to the more challenging code generation task. Specifically, CodeT5Mix already achieved competitive generative results (20.47 BLEU4) compared to the retrieval-augmented generative results of REDCODER (21.01 BLEU4) on Python. We will include this experiment in our final revision.

---

> > ### Author Response · Authors · 2022-11-14
> > **Response to Reviewer-eBFF (2/n)**
> >
> > ### Q4: “What would happen if you trained a single encoder-decoder with all the training objectives you have on the same pretraining dataset you have? I think you have shown something very close in the ablation, but the options in ablation are not clear.”
> > We make the ablation study (Section 4.4) clearer with more explanations in the revised version. Indeed, we did have an ablation result using a single encoder-decoder (the “shared decoder” variant in the last row of Table 6). We found that this variant consistently yields lower performance compared to CodeT5Mix on text-to-code retrieval, code summarization and generation, which might be due to the inter-task interference as the model relies on a single model for multiple different tasks. By contrast, the weight-sharing design in CodeT5Mix can reduce such interference without significant extra parameter cost, i.e. only two additional FFN task-experts. Such extra parameter cost is further waived when applied to a specific task as we only need to activate a single corresponding FFN task-expert.
> > ### Q5: “What is the justification for text-code matching decoder? Can't you do that task using the [CLS] token representation from encoder?”
> > The benefits of having a text-code matching decoder are two-fold:
> >
> > - This strategy provides an additional way to adapt current models to downstream understanding tasks. The first way is to employ the bimodal encoder to obtain text/code representations and pass them to a classification layer. Alternatively, we can additionally use the text-code matching decoder to learn a better text-code bimodal representation, which is critical for downstream tasks such as text-to-code retrieval. From the ablation study in Table 6, we can confirm its significance with an average performance drop of 1.65 absolute MRR score over 6 text-to-code retrieval tasks after removing the matching decoder.
> > - This strategy can efficiently activate common components of the decoder across a diverse set of cross-modal pretraining tasks. Specifically, except for the feed-forward layers, the text-code matching decoder shares the same parameters of the causal self-attention and cross-attention layers with other decoders (i.e. code generation and text generation decoders). This weight-sharing scheme allows cross-task generalization whereby the contextual representations in attention layers learned from text-code matching tasks can complement other tasks such as code/text generation tasks. From Table 6, we can observe that removing the matching decoder decreases the performance of other tasks such as code summarization and generation.
> >
> > ### Q6: “Why two stage training? …what is the point of using those two CLM in the 2nd stage, those tasks are already sufficiently trained in the first stage?”
> > The two-stage pretraining strategy is designed based on two different pretraining datasets: large-scale code-only data and smaller-scale text-code data. Our motivation is to tradeoff between pretraining efficiency and data scales:
> > - In the first stage, we focus on unimodal code-only pretraining data which is usually available in large amounts (e.g. Github data). To pretrain on this data, we only need to focus on pretraining tasks such as span denoising and causal language modeling. These learning objectives are easy to train and do not require any annotation efforts. Note that it is possible to adopt contrastive and matching losses in this stage too by considering pairs of code from the same program. However, we expect these types of pretraining setups are less useful to downstream tasks, which was confirmed through our conducted preliminary experiments.
> > - In the second stage, we focus on parallel text-code pretraining data which is usually available with limited annotations (e.g. CodeSearchNet). In this stage, we focus on more complex pretraining objectives, including contrastive learning and matching losses. Variants of CLM tasks such as code-to-text and text-to-code generation are also adopted to further adapt the model to downstream multimodal tasks. Note that these two CLMs are different from the first stage which only focuses on code generation without clear boundaries between text and code sequences.
> > ### Q7: “How are the losses combined in the second stage?”
> > As we mentioned in the “Pretraining Setup” paragraph (original paper), we jointly optimize four losses of contrastive learning, matching, and two CLM losses for 10 epochs. We combine these four losses with equal weights. In the revised paper, such information can be also found in Section 3.2 and more pretraining details can be found in Appendix B.2.

---

> > > ### Comment · Reviewer_eBFF · 2022-11-19
> > > **Further discussion**
> > >
> > > EDIT: Thanks for answering and clearing up some of the confusions I had based on the earlier version of the paper.
> > >
> > > One concern still remains about the results of retrieval augmented summarization hampering unequivocal supremacy of CodeT5Mix.
> > > Nevertheless, we cannot hope for one model to solve all the problems. Hence, I have updated my score based on current version of the paper.

---

### Author Response · Authors · 2022-11-14
**Response to all Reviewers**

We thank all the reviewers for their insightful feedback! We are glad that all reviewers agree that our submission has a thorough evaluation with extensive analysis and the paper is well written. We are encouraged that reviewers find our method novel (R1-eBFF, R3-ZFgG, R4-3yBT) and our idea interesting (R1-eBFF, R2-EgG1), our model performance strong and impressive (R5-stVy, R1-eBFF), and our overall paper with high quality (R4-3yBT).

We updated a new version of the paper based on reviews to resolve several issues, including clearer description of the model architecture and pretraining tasks, elaborating more on the motivation of the matching decoder, better explanation of the ablation study, more experimental analysis, etc.

### Response to the novelty concern
We want to emphasize that the novelty of our work is not about the individual pretraining tasks/objectives, but a unified pretraining framework ( i.e. “CodeT5Mix” ) that cohesively integrates multiple pretraining tasks with a purposeful weight-sharing design among task-specific experts. Our approach enjoys the benefits of multi-task joint training while less affected by any inter-task interference. Typically, CodeT5Mix is novel from two perspectives.

- From a model architecture perspective, we propose a flexible mixture architecture to operate in various modes (encoder-only, decoder-only, and encoder-decoder) to suit different downstream tasks. This is different from the existing code pretrained models architectures which are based either BERT-style or GPT-style or T5-style.
- From a learning objective perspective, we propose a mixture of self-supervised objectives to endow the model with versatile code understanding and generation capabilities. Although each of the individual objectives is not new and can be found in prior LMs in the domain of either NLP or CV (computer vision), we argue that it is novel to propose the unified method that can effectively combine them together to build a powerful LM that excels in a wide range of downstream tasks in the code domain.
### Response to the model performance concern

We conduct comprehensive evaluations (as recognized by all reviewers) in *7 different code-related tasks over 20+ datasets*. We show that CodeT5Mix achieves new SoTA performance with significant performance gain in most cases, including:
- 8 text-to-code retrieval tasks: +3.16 avg. MRR than prior SoTA UniXcoder, +5.88 avg. MRR than CodeT5.
- 2 line-level code completion tasks: +1.74 EM than UniXcoder, +7.89 EM than CodeT5 on PY150, and +3.37 EM than UniXcoder, +11.47 EM than CodeT5 on JavaCorpus.
- 2 math programming tasks: +2.7 Pass@80 than GPT-Neo and +15.1 Pass@80 on MathQA-Python, +2.8 Pass@100 than Codex and +12.4 than CodeT5 on GSM8K-Python.
- 2 code generation tasks in retrieval-augmented generation setting:  +4.85 and +6.71 BLEU-4 than prior SoTA REDCODER-EXT on Java and Python datasets respectively.

Note that for other tasks such as code defect detection and clone detection tasks, we can find that their performance often becomes saturated (reflected by smaller performance gaps between the top-performing models), where our models are still better than or on par with the SoTA models.

**Extensive Ablation study.** We conduct a comprehensive ablation study to validate the mixture architecture design (see the middle block in Table 6) and weight sharing strategy (see the last block in Table 6) in *3 representative tasks over 13 datasets*. These ablation results show that the performance gains from CodeT5Mix components are significant and consistent in many cases. Please refer to Section 4.4 in the revised paper for more details.

Once more, thank all the reviewers for the insightful feedback and we are looking forward to the discussion.

---

### Author Response · Authors · 2022-11-17
**Response to all Reviewers**

Thanks all the reviewers for the discussion/feedback to our paper! As the discussion period is ending, please let us know if you have any additional questions or concerns that still block you from increasing the recommendation scores. We would be very happy to address them.

---

### Decision · Program_Chairs · 2023-01-20

**Decision:**

Reject

**Justification For Why Not Higher Score:**

It is possible that a reader generalizes the findings of this work in a way that is not warranted. I believe that for this paper to be published in ICLR the issues in its evaluation need to be addressed first.

**Justification For Why Not Lower Score:**

N/A

**Metareview: Summary, Strengths And Weaknesses:**

This paper makes the following contributions:
- Presents a pretraining strategy for transformer-based models of code.
- An encoder-decoder Transformer model where different tasks share all model parameters apart from the feed-forward layers of the transformer.
- An empirical evaluation over a large range of tasks.

## Strengths and Weaknesses
(+) Substantial experiments covering a wide variety of different tasks.

(+) An interesting set of ideas wrt to weight sharing and pretraining objectives

(-) Empirical comparison often conflates the advantages one gets from the different pretraining objective, the weight sharing/separation, and the model architecture. Specifically:

* the total amount of compute and parameters is not always comparable as the task-specific experts regime introduces more weights that could have been used in a standard transformer architecture (e.g. to add an additional layer or increase the hidden size).
* Similarly, the evaluation seems to conflate the (dis)advantages of an encoder-decoder architecture, with decoder-only, or UniLM-style (UniXCoder) and pretraining objectives and weight sharing. In particular, the similar pretraining objectives and weight sharing could be used with decoder-only or Uni-LM-style models. Would the advantages observed in CodeT5Mix transfer to those models too (e.g., if weight sharing is introduced or the same pretraining objective is used)?


Given the above, I suggest to not to accept this paper at its current state. While it represents a substantial amount of work (and compute), the evaluation needs to be refined to ensure that the contributions of this work generalize effectively and can be used by others.


**Summary Of Ac-Reviewer Meeting:**

While the scores originally were borderline, in an offline discussion the reviewers agreed to reject this paper at its current state.